# Giant second harmonic transport under time-reversal symmetry in a trigonal superconductor

Yuki M. Itahashi[1], Toshiya Ideue[1 ✉], Shintaro Hoshino[2], Chihiro Goto[1], Hiromasa Namiki[3], Takao Sasagawa[3] & Yoshihiro Iwasa[1,4]

Nonreciprocal or even-order nonlinear responses in symmetry-broken systems are powerful probes of emergent properties in quantum materials, including superconductors, magnets, and topological materials. Recently, vortex matter has been recognized as a key ingredient of giant nonlinear responses in superconductors with broken inversion symmetry. However, nonlinear effects have been probed as excess voltage only under broken time-reversal symmetry. In this study, we report second harmonic transport under time-reversal symmetry in the noncentrosymmetric trigonal superconductor $PbTaSe_2$. The magnitude of anomalous nonlinear transport is two orders of magnitude larger than those in the normal state, and the directional dependence of nonlinear signals are fully consistent with crystal symmetry. The enhanced nonlinearity is semiquantitatively explained by the asymmetric Hall effect of vortex-antivortex string pairs in noncentrosymmetric systems. This study enriches the literature on nonlinear phenomena by elucidating quantum transport in noncentrosymmetric superconductors.

[1] Quantum-Phase Electronics Center (QPEC) and Department of Applied Physics, The University of Tokyo, Tokyo 113-8656, Japan. [2] Department of Physics, Saitama University, Saitama 338-8570, Japan. [3] Laboratory for Materials and Structures, Tokyo Institute of Technology, Kanagawa 226-8503, Japan. [4] RIKEN Center for Emergent Matter Science (CEMS), Wako 351-0198, Japan. ✉email: ideue@ap.t.u-tokyo.ac.jp

Recently, symmetry breaking in solids has become the focus of research in condensed matter physics. It is also a key strategy for developing novel functionalities. To date, many characteristic physical properties, which are unique to non-centrosymmetric crystals, have been reported. For example, the nonlinear optical response such as the second harmonic generation and optical parametric effect are known to occur in non-centrosymmetric crystals[1]. Broken inversion symmetry also affects transport properties via asymmetric scattering, spin–orbit interaction, magnetic structure, and accompanying geometrical/topological characteristics[2,3].

Among the various emergent transports originating from symmetry breaking, the second-order nonlinear transport, which includes the intrinsic rectification effect and nonlinear Hall effect, is recognized as a sophisticated probe of symmetry breaking and a potential functionality for rectifying a variety of quantum currents[4–25]. To date, it has been studied mainly in systems with broken time-reversal symmetry[4–17]. Recently, however, it has been proposed that second-order nonlinear transport can occur even under time-reversal symmetric conditions. An important example is the nonlinear anomalous Hall effect[21], which is a new type of Hall effect realized under time-reversal symmetric conditions and has been experimentally observed in few-layer WTe$_2$[22,23] and bulk TaIrTe$_4$[24]. Band topology/geometry (i.e., Berry curvature dipole)[22] and anomalous scattering (skew-scattering-like mechanism)[23] have been reported to be the origins of the nonlinear Hall effect. However, no occurrence of the intrinsic rectification effect in the longitudinal resistance under time-reversal symmetric conditions has been reported so far, and it is necessary to investigate more materials that show the anomalous nonlinear transverse response with a distinctive origin. Furthermore, the search for anomalous nonlinear transport in exotic quantum states, such as superconductivity, is a significant challenge from both fundamental and technological points of view.

In this study, we investigated the second-order nonlinear anomalous transport in PbTaSe$_2$, which is a noncentrosymmetric trigonal superconductor and has attracted increased interest as a possible topological material[26–31]. We observed a second-order nonlinear transverse response, as well as a rectification effect that satisfies the characteristic directional dependence for the trigonal symmetry. Remarkably, we found that the nonlinear anomalous transport under the time-reversal symmetric condition was enhanced by orders of magnitude in the superconducting (SC) state. Second-order nonlinear transport exhibited a peak structure around the superconducting transition, indicating that vortices/antivortices excited in the layered material played a major role in nonlinear transport. The observed superconducting nonlinear transverse response/rectification effect can be explained by the asymmetric Hall effect of vortices and antivortices owing to the rectification by trigonal potentials.

## Results

**Basic properties of PbTaSe$_2$.** PbTaSe$_2$ is composed of alternating stacking of 1H-TaSe$_2$ and Pb layers (Fig. 1). Because each 1H-TaSe$_2$ layer has a noncentrosymmetric trigonal structure (Fig. 1a) and the stacking direction is the same for all the TaSe$_2$ layers (Fig. 1b), multilayer PbTaSe$_2$ also has a noncentrosymmetric trigonal symmetry[26–28]. Furthermore, it exhibits a superconducting transition at $T_c = 3.7$–$3.8$ K[22,23]; therefore, it is an ideal platform for investigating second-order nonlinear transport in superconducting states.

Thin flake samples are highly beneficial for the study of nonlinear transport because a large current density is easily obtained. Therefore, we fabricated micro-size PbTaSe$_2$ devices (Fig. 1c) with typical thicknesses of approximately 100 nm

through the exfoliation method. We prepared samples with two different configurations (Fig. 2a): the current flowing along a zigzag direction (configuration A: samples 1, 3, and 6) and armchair direction (configuration B: samples 2, 4, 5, and 7). The crystal orientation was well identified by the flake shapes[32] and scanning transmission electron microscopy (STEM) measurements. For example, in Fig. 1d, we depict the STEM image of the cross-section along the black dashed line in Fig. 1c. The STEM image is identical to the cross-sectional image of PbTaSe$_2$ along the armchair direction (inset in Fig. 1d), indicating that the current flows in the zigzag direction in sample 1. Figure 1e depicts the temperature variations of first harmonic resistance $R_{xx}^{\omega}$ in sample 2. It indicates a metallic behavior with a residual resistivity ratio value of ~100, which is consistent with the values reported in previous studies on bulk crystals[27–29]. Figure 1f depicts a magnified view of the temperature dependence of $R_{xx}^{\omega}$ around the superconducting transition. The superconducting transition temperature, $T_c$, defined as the midpoint of the resistive transition ($B = 0$ T, red line) is 3.6 K, which is also consistent with previous studies on bulk crystals[26,27]. When a magnetic field of 1 T was applied (orange line), the superconducting transition was completely suppressed. In the following, we discuss transport only under the time-reversal symmetric condition, that is, without a magnetic field, unless stated otherwise.

**First and second harmonic resistance in the normal state.** First, we focus on the second harmonic resistance in the normal state. In trigonal crystals, a second-order nonlinear voltage can appear along the armchair direction when current is applied along either the armchair or zigzag direction (see "Methods"). The intrinsic rectification effect (nonlinear transverse response) is expected in response to the applied current along the armchair (zigzag) direction (Fig. 2a). A recent theory proposed second-order nonlinear transport in trigonal systems under time-reversal symmetry[33] as well as a possible mechanism. Note that a trigonal crystal has three mirror planes and is thus nonpolar. This is in contrast to WTe$_2$ with only one mirror plane, along which the second-order nonlinear transverse voltage has been observed[22,23]. In trigonal crystals, the effect of the Berry curvature dipole can be eliminated owing to the high symmetry. Therefore, we can investigate other possible origins such as skew scattering or unknown effects including a higher-order Berry curvature distribution in momentum space.

Figure 2b, c depict the current dependences of second harmonic resistance ($R^{2\omega}$) in configuration A (sample 3, $T = 20$ K) and configuration B (sample 4, $T = 50$ K), respectively. When $I$ is applied along the zigzag (armchair) direction, the finite $I$-linear $R_{yx}^{2\omega}$ ($R_{xx}^{2\omega}$) is observed, which is significantly larger than $R_{xx}^{2\omega}$ ($R_{yx}^{2\omega}$). These results are consistent with the above symmetry argument for the threefold rotational symmetry (see "Methods"), which unambiguously excludes the possibility of unexpected nonlinear responses coming from extrinsic effects such as asymmetric shapes and/or configurations of electrodes. We note that similar nonlinear transverse response has been reported in the trigonal surface of Bi$_2$Se$_3$[25]. In the present PbTaSe$_2$, the intrinsic rectification effect (Fig. 2c), which has never been reported under time-reversal symmetry, is also clearly observed as well as the nonlinear transverse response (Fig. 2b).

To further understand the nonlinear transverse response in the normal state, we measured the temperature variations of the normalized nonlinear transverse signal $\frac{|E_y^{(2)}|}{(E_x^{(1)})^2}$ (see "Methods") and linear conductivity $\sigma_{xx}^{\omega} = \left(\frac{Wt}{L} R_{xx}^{\omega}\right)^{-1}$ of sample 1 are depicted in Fig. 2d. Here, $W = 3.5$ μm, $L = 1.7$ μm, and $t = 123$ nm are the sample width, distance between electrodes, and sample thickness,

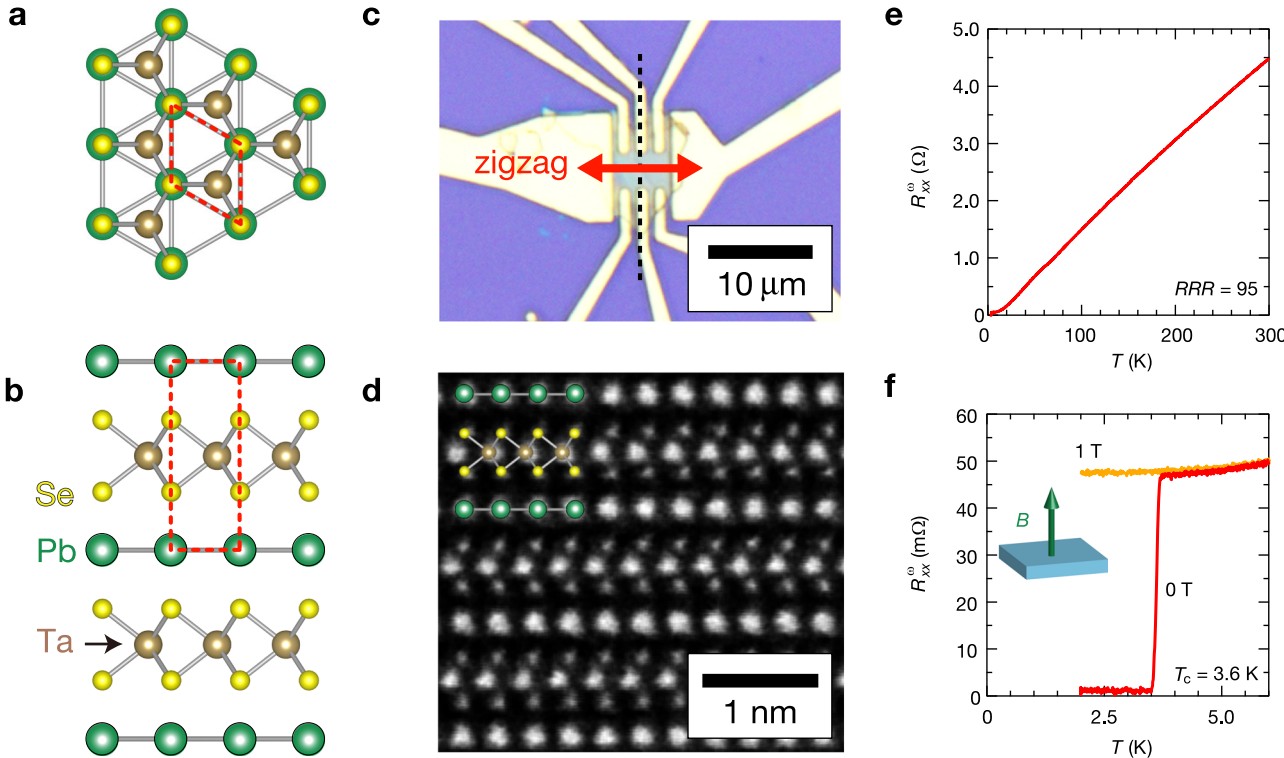

**Fig. 1 Crystal structure and superconducting properties of PbTaSe₂.** (**a**, **b**) **a** Top and **b** side views of PbTaSe₂. Pb layers are intercalated in TaSe₂ with 1H stacking. Red dashed squares indicate the unit cell. **c** Optical microscope image of the PbTaSe₂ device (sample 1). **d** A cross-sectional scanning transmission electron microscope (STEM) image of PbTaSe₂ along the dashed line in (**c**). Schematic of the cross section of PbTaSe₂ along the armchair direction is also displayed. **e** Temperature dependence of the first harmonic resistance $R_{xx}^{\omega}$ in sample 2. **f** Temperature dependence of $R_{xx}^{\omega}$ around the superconducting transition ($T_c = 3.6$ K) in sample 2 when $I = 140$ μA. Red and orange lines depict the data under 0 T (red) and 1 T (orange), respectively. Magnetic field $B$ is applied perpendicular to the 2D layers.

respectively. Both $\frac{|E_y^{(2)}|}{(E_x^{(1)})^2}$ and $\sigma_{xx}^{\omega}$ increased monotonically with a decrease in the temperature, exhibiting similar behavior. In Fig. 2e, we analyze the correlation between these quantities by plotting $\frac{|E_y^{(2)}|}{(E_x^{(1)})^2}$ versus $(\sigma_{xx}^{\omega})^2$. $\frac{|E_y^{(2)}|}{(E_x^{(1)})^2}$ indicates a linear dependence on $(\sigma_{xx}^{\omega})^2$, particularly in the high $\sigma_{xx}^{\omega}$ (low-temperature) region. Unexpectedly, a nonlinear anomalous response was visible even above $T = 100$ K. In general, the second-order nonlinear transverse voltage can be well described by equation $\frac{|E_y^{(2)}|}{(E_x^{(1)})^2} = \xi(\sigma_{xx}^{\omega})^2 + \eta$ (where $\xi$ and $\eta$ are phenomenological fitting parameters)[21], reflecting the two contributions to the nonlinear transverse response: the first term can be the skew-scattering-like origin, which scales as $\frac{|E_y^{(2)}|}{(E_x^{(1)})^2} \propto \tau^2 \propto (\sigma_{xx}^{\omega})^2$, and the second term that satisfies $\frac{|E_y^{(2)}|}{(E_x^{(1)})^2} \propto \tau^0 \propto (\sigma_{xx}^{\omega})^0$ corresponds to scattering-time-free mechanisms such as the Berry curvature dipole effect and the side-jump mechanism. In sample 1, fitting parameters $\xi$ and $\eta$ are estimated as $\xi = 2.3 \times 10^{-20}$ m³ V⁻¹ Ω² and $\eta = -3.2$ μmV⁻¹, respectively. Unlike a previous study on WTe₂[23], in which both contributions cannot be neglected, the first term is dominant in PbTaSe₂. This might be because the Berry curvature dipole will strictly vanish in trigonal crystals with three mirror planes. Note that a small deviation from the relation $\frac{|E_y^{(2)}|}{(E_x^{(1)})^2} \propto \tau^2 \propto (\sigma_{xx}^{\omega})^2$ (black linear dashed line in Fig. 2e) was observed in the low $\sigma_{xx}^{\omega}$ (high-temperature) region. This might be attributed to the contribution from the $\sigma_{xx}^{\omega}$-linear term ($\frac{|E_y^{(2)}|}{(E_x^{(1)})^2} \propto \sigma_{xx}^{\omega}$) originating from both

skew and side-jump scatterings[34]. The same scaling of $\frac{|E_y^{(2)}|}{(E_x^{(1)})^2}$ and $(\sigma_{xx}^{\omega})^2$ was observed in other samples, as depicted for sample 3 in Supplementary Fig. 5.

**First and second harmonic resistance in the superconducting state.** Next, we focus on nonlinear transport in the SC state. Figure 3a, b depict the current dependences of $R^{2\omega}$ (left) and $R_{xx}^{\omega}$ (right) in configurations A (sample 1) and B (sample 2), respectively, at $T = 2$ K. With an increase in the current, the superconducting zero-resistance state was broken and a finite resistance state appeared (black dotted curve). Around this transition, a sharp peak of $R_{yx}^{2\omega}$ ($R_{xx}^{2\omega}$) was observed when $I$ was applied parallel to the zigzag (armchair) direction. Note that such anomalies are negligibly small in other directions, in fair agreement with the directional dependence of second-order nonlinear transport in the trigonal systems, as in the case of the normal state (Fig. 2b, c). Figure 3c, d depict the temperature dependences of $R^{2\omega}$ (left) and $R_{xx}^{\omega}$ (right) in configurations A (sample 1, $I = 0.06$ mA) and B (sample 2, $I = 0.3$ mA), respectively. A peak behavior similar to Fig. 3a, b was observed in $R_{yx}^{2\omega}$ ($R_{xx}^{2\omega}$), whereas such a signal was small or absent in the other direction when $I$ was parallel to the zigzag (armchair) direction. Figure 3a–d indicate that both the nonlinear transverse response and the rectification effect were significantly enhanced in the transition region and suppressed in the zero-resistance state. Such nonlinear anomalous transport, which satisfies the directional dependence of trigonal symmetry and is enhanced in the SC fluctuation region, was observed in all the samples we measured (Supplementary Note 5).

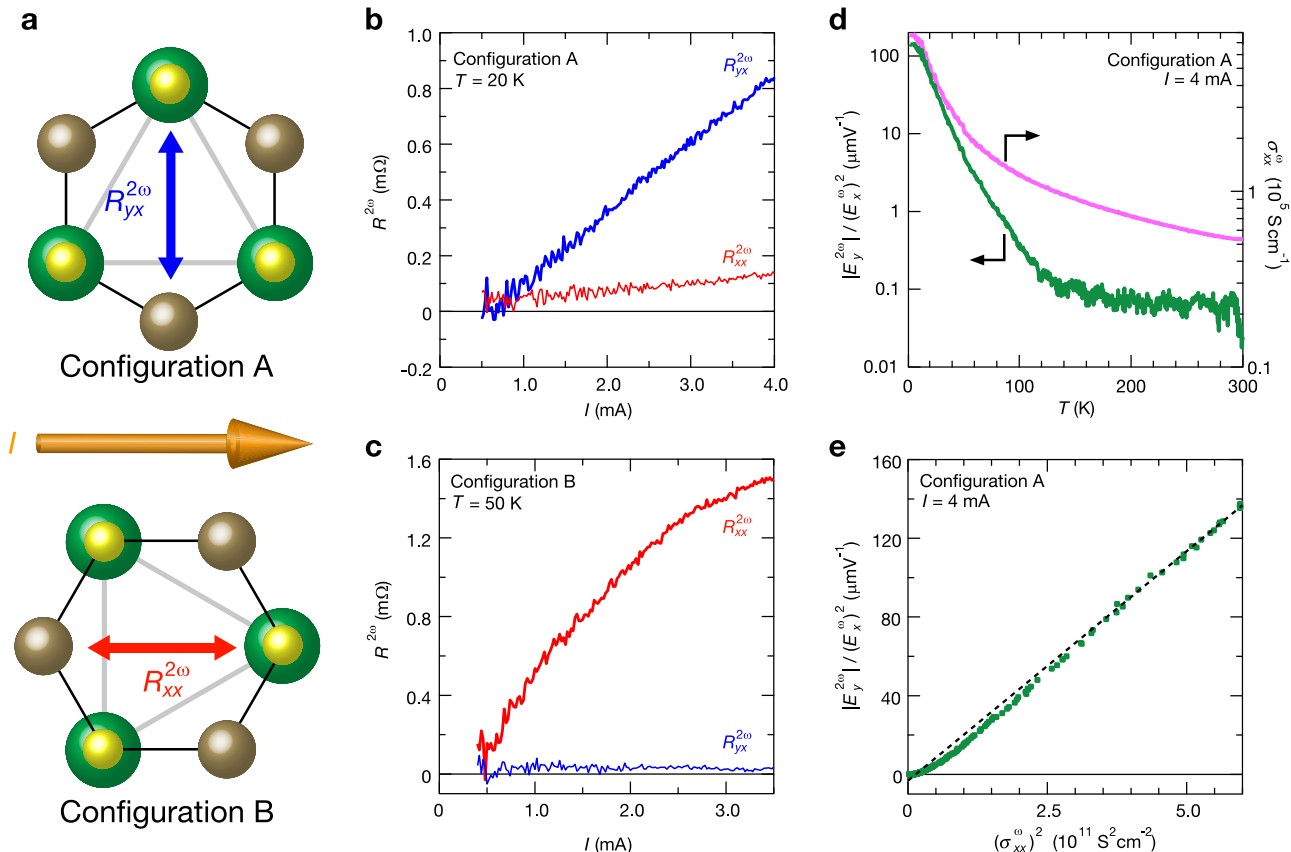

**Fig. 2 Current and temperature dependences of second-order anomalous transport in normal state. a** Schematics of crystal structure and expected directional dependence (selection rule) of the nonlinear transverse response/rectification effect in trigonal PbTaSe$_2$. In configuration A (B), where the current is applied along the zigzag direction (armchair direction), the second harmonic signal is expected in the transverse (longitudinal) direction. **b**, **c** Current dependences of the second harmonic resistance $R^{2\omega}$ in **b** configurations A (sample 3) at $T = 20$ K and **c** configuration B (sample 4) at $T = 50$ K. Red and blue lines indicate longitudinal ($R_{xx}^{2\omega}$) and transverse ($R_{yx}^{2\omega}$) resistance, respectively. Directional dependence of the nonlinear transport illustrated in Fig. 2a is confirmed. **d** Normalized second harmonic response $\frac{|E_y^{(2)}|}{(E_x^{(1)})^2}$ (green, left) and first harmonic conductivity $\sigma_{xx}^{\omega}$ (pink, right) as a function of temperature at $I = 4$ mA in configuration A (sample 1). **e** Normalized second harmonic resistivity $\frac{|E_y^{(2)}|}{(E_x^{(1)})^2}$ as a function of $(\sigma_{xx}^{\omega})^2$ in configuration A (sample 1). Black dotted line indicates the linear fitting in the low-temperature (high-conductivity) region.

Around the superconducting transition, excited vortex–antivortex pairs or vortex loops are known to cause a resistive state in 2D or layered superconductors even under the time-reversal symmetric condition[35,36]. In the present case of layered PbTaSe$_2$, our simulation revealed that the vortex–antivortex string pair had the lowest energy excitation, as described in Supplementary Note 3. Therefore, the system can be regarded as 2D from the vortex point of view. We propose that this vortex/antivortex dynamics causes the nonlinear transverse response during the superconducting transition, as discussed below, in a manner similar to the vortex rectification effect in trigonal superconductors under an out-of-plane magnetic field[15,18]. Although we note another possible contribution from the amplitude fluctuations above the superconducting transition temperature, we mainly focus on the vortex/antivortex dynamics in this work because it will be dominant below the transition temperature and the following theoretical model can also semiquantitatively explain the results.

In Fig. 3e, we depict a possible mechanism for the observed nonlinear transverse voltage in trigonal superconductors by considering the asymmetric vortex/antivortex Hall effect owing to the trigonal potential. The first clue came from the observation of the excess component in the Hall resistance, which was interpreted as a vortex Hall effect[37–44] (Supplementary Note 2 and Supplementary Figs. 1, 2). The origin of the vortex Hall effect is still being debated. One potential mechanism is the charging of

the vortex core due to the difference between the chemical potentials of the normal core and superconducting states. We consider that the vortices and antivortices are excited by a finite temperature or current as string pairs even without magnetic fields (see Supplementary Note 3, Supplementary Fig. 3). When current is applied along the zigzag direction (configuration A), vortices/antivortices are first driven in the armchair direction and then curved in the transverse zigzag direction owing to the vortex/antivortex Hall effect. During this process, vortices/antivortices are rectified, reflecting the trigonal potential; therefore, the vortex Hall effect is asymmetric. This results in the antiparallel motion of vortices and antivortices, which is equivalent to the net flow of vorticity current (purple arrow) in Fig. 3e; the excess voltage appears perpendicular to it, or along the armchair direction, and is observed as the nonlinear transverse voltage. A similar scenario also explains the intrinsic rectification effect (see Supplementary Note 3, Supplementary Fig. 3).

This model is formulated in Supplementary Note 4. In this theoretical description, we consider the vortex/antivortex dynamics, particularly the Hall effect in trigonal potentials (Supplementary Eq. 5). By combining the rectification effect and Hall effect of vortices/antivortices[18], we obtained the expression of $R_{yx}^{2\omega}$ as $R_{yx}^{2\omega} = \frac{(\phi_0^*)^3 n_v r \ell_v I}{k_B T W \eta_0} g_2(\frac{U}{k_B T})$, where $\phi_0^*$ is the flux quanta, $n_v$ is the total number density of the vortices/antivortices,

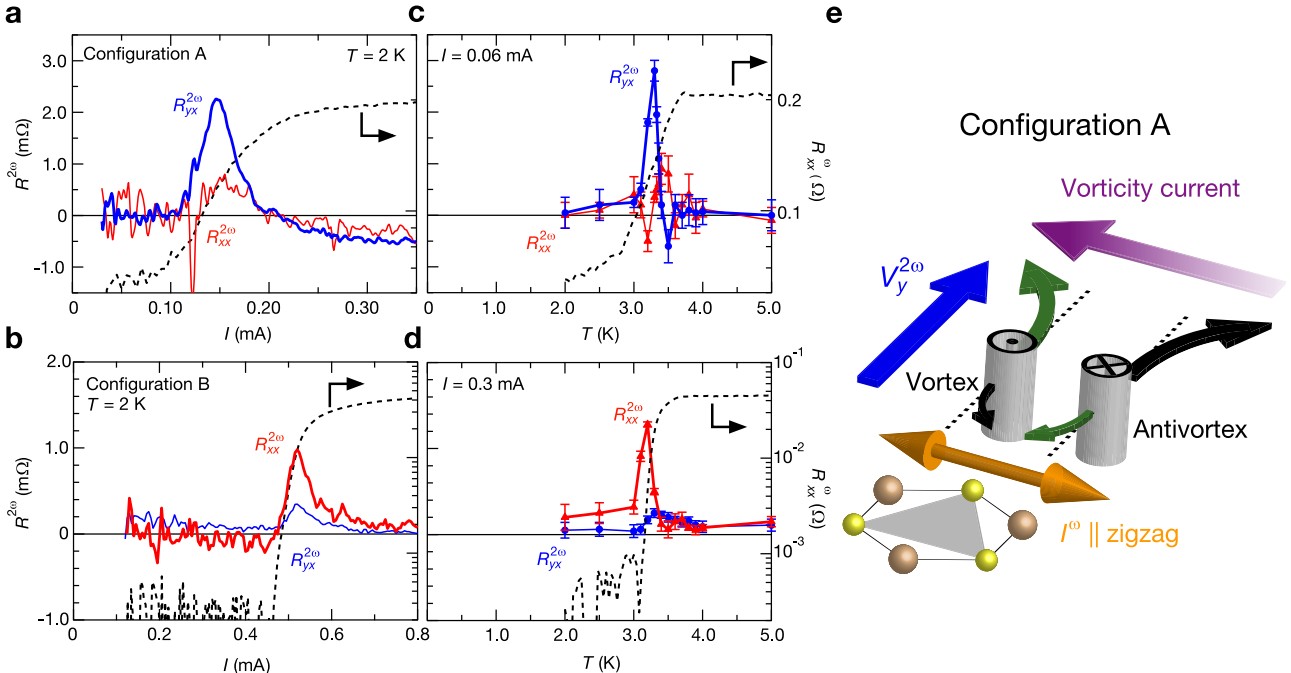

**Fig. 3 Current and temperature dependences of first/second harmonic signals around superconducting transition and schematic of asymmetric vortex Hall effect as a possible origin for nonlinear transport. a, b** Current dependence of $R^{2\omega}$ (left) and $R_{xx}^{\omega}$ (right) at $T = 2$ K in **a** configuration A (sample 1) and **b** configuration B (sample 2). Red and blue lines indicate longitudinal ($R_{xx}^{2\omega}$) and transverse ($R_{yx}^{2\omega}$) resistance, respectively. **c, d** Temperature dependences of $R^{2\omega}$ (left) and $R_{xx}^{\omega}$ (right) in **c** configuration A (sample 1) and **d** configuration B (sample 2). The current value is 0.06 mA and 0.3 mA in **c, d**, respectively. Red and blue lines indicate longitudinal ($R_{xx}^{2\omega}$) and transverse ($R_{yx}^{2\omega}$) resistances, respectively. Errorbars indicate the uncertainty of the signals estimated from current dependence of the second harmonic signals at each temperature. **e** Schematic of the rectified vortex/antivortex Hall effect in configuration A. Black (green) arrows denote the trajectories of the vortices/antivortices Hall effect when current flows along the zigzag direction. The rectification of vortices/antivortices reflecting the trigonal potential is represented by the thickness differences of arrows. Purple arrow indicates the antiparallel motions of vortices or vorticity current. Nonlinear voltage ($V_y^{2\omega}$) appears perpendicular to the vorticity current, in analogy to the inverse spin Hall effect. Regardless of the direction of the current, excess voltage with the same sign appears along the armchair direction, which can be observed as the second harmonic resistance.

$r$ is the Hall angle of vortices, and $\eta_0$ is the friction coefficient. Parameters $\ell_v$, $U$, and $g_2$ are length, energy, and dimensionless function, respectively, which are determined from the detailed shape of the asymmetric pinning potentials by using Fokker–Planck equation (Supplementary Eq. 9). In this study, we assumed that the dissociation of vortex–antivortex pairs is induced predominantly by the current injected for the observation of the nonlinear transport effect. We employed the realistic phenomenological parameters: The potential $U$ and the length $\ell_v$ are determined from the experimental current density and magnetic field where the vortices are depinned. $n_v$ is determined from the maximum number of vortices which can be excited in the sample because nonlinear responses are generated by thermal or current-noise fluctuations near the transition point (Supplementary Note 4). Thus, we estimated the value of $R_{yx}^{2\omega}$ to be approximately 1.4 m$\Omega$ near the superconducting transition temperature (Supplementary Fig. 4b), which is in good agreement with the experimental results. Our theoretical model can also explain the temperature dependence of $R_{yx}^{2\omega}$ in the superconducting region and the magnitude difference of nonlinear transport between the superconducting and normal states (see Supplementary Note 4). The detailed analysis on the vortex pinning profile and superconducting fluctuation effect will further clarify the nature of nonreciprocal signals.

In the Supplementary Note 6, we also discuss the nonreciprocal transport under a magnetic field[10–17] to obtain a comprehensive understanding of the vortex dynamics in this material (Supplementary Fig. 6). The directional dependence of the antisymmetric

second-order nonlinear magnetoresistance is rotated by 90° from the case under time-reversal symmetry, which further supports the intrinsic nature of the signals. Significantly, the theoretical estimation of the magnitude of the nonlinear magnetotransport is consistent with the experimental results, as explained in Supplementary Note 4. This result also supports the above scenario, based on the asymmetric vortex Hall effect.

## Discussion

In Fig. 4a, we compare the nonlinear transverse signals in the normal and SC states. The temperature dependence of $\frac{|E_y^{(2)}|}{(E_x^{(1)})^2}$ in both the normal state (blue; $I = 4.3$ mA) and the SC state (red; $I = 100\ \mu$A) (left) are plotted as well as $R_{xx}^{\omega}$ at $I = 100\ \mu$A (right). Note that superconductivity is destroyed even below $T_c$ when a large current ($I = 4.3$ mA) is applied. The obtained values of $\frac{|E_y^{(2)}|}{(E_x^{(1)})^2}$ below $T_c$ are smoothly connected to the normal state contribution. $\frac{|E_y^{(2)}|}{(E_x^{(1)})^2}$ in the SC state at $I = 100\ \mu$A indicates a remarkable enhancement by orders of magnitude compared to that in the normal state. A similar gigantic enhancement of second-order nonlinear transport is also observed in the nonreciprocal magnetotransport[13–15,17], implying that nonlinear transport is universally enhanced in the SC state, regardless of the time-reversal symmetry being preserved or not.

Finally, we compare the nonlinear transverse signals observed in the present system of PbTaSe$_2$ and with those previously reported for few-layer WTe$_2$[22,23], TaIrTe$_4$[24], and Bi$_2$Se$_3$ surface[25].

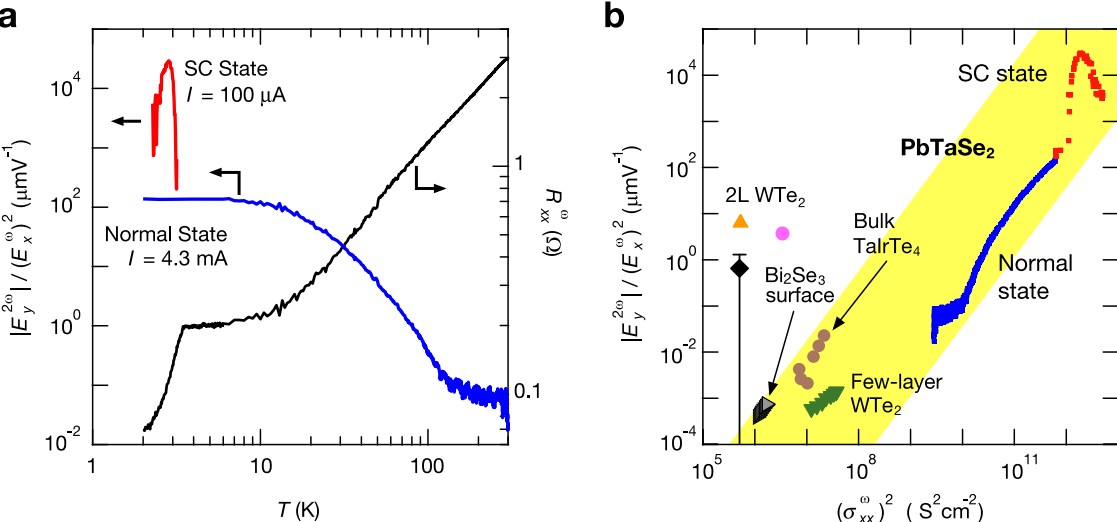

**Fig. 4 Summarized temperature dependence of nonlinear transverse response in PbTaSe₂ and nonlinear transverse responses in various materials.**
**a** Temperature dependences of $\frac{|E_y^{(2)}|}{(E_x^{(1)})^2}$ in the normal state (blue; $I = 4.3$ mA) and in the superconducting (SC) state (red; $I = 100$ μA) (left), and $R_{xx}^\omega$ at $I = 100$ μA (right) in sample 1. **b** $\frac{|E_y^{(2)}|}{(E_x^{(1)})^2}$ as a function of $(\sigma_{xx}^\omega)^2$ in PbTaSe₂, few-layer WTe₂[22,23], TaIrTe₄[24], and Bi₂Se₃ surface[25]. Blue and red squares indicate $\frac{|E_y^{(2)}|}{(E_x^{(1)})^2}$ of PbTaSe₂ (sample 1) in the normal state and the SC state, respectively. Green triangles depict $\frac{|E_y^{(2)}|}{(E_x^{(1)})^2}$ in the few-layer WTe₂. Data were sourced from Kang et al.[23]. Black diamond, pink circle, and orange triangle represent $\frac{|E_y^{(2)}|}{(E_x^{(1)})^2}$ in the bilayer (2 L) WTe₂. These were calculated using data from Ma et al.[22]. Carrier densities $n$ are approximately 0 cm⁻² (black), $-7 \times 10^{12}$ cm⁻² (pink), and $-7 \times 10^{11}$ cm⁻² (orange) ($n$ values corresponding to carrier densities where the nonlinear transverse response exhibits the local maximum values). The errorbar indicates the sign change in $\frac{|E_y^{(2)}|}{(E_x^{(1)})^2}$. Brown circles depict $\frac{|E_y^{(2)}|}{(E_x^{(1)})^2}$ in the 16 nm-thick TaIrTe₄. Data were sourced from Kumar et al.[24]. Gray triangles depict $\frac{|E_y^{(2)}|}{(E_x^{(1)})^2}$ in the Bi₂Se₃ surface. Data were sourced from He et al.[25].

In Fig. 4b, the values of $\frac{|E_y^{(2)}|}{(E_x^{(1)})^2}$ are plotted as a function of $(\sigma_{xx}^\omega)^2$ for all materials. Similar plots of anomalous transverse signal versus longitudinal conductivity are known to be useful for discussing the mechanisms of the linear anomalous Hall effect in itinerant magnets[45] and anomalous thermal Hall effect in insulators[46]. In bilayer (2 L) WTe₂, the conductivity is small because the Fermi level is located near the band edge and the behavior of the nonlinear transverse response is rather complex, even exhibiting a sign change depending on the Fermi level position and the electrical displacement field. The observed nonlinear transverse signal can be explained well by the Berry curvature dipole effect in this case[22]. In few-layer WTe₂ and TaIrTe₄, conductivity increases and both the skew scattering mechanism and the Berry curvature dipole effect are discussed as the origin of the nonlinear transverse response, which has already been discussed in the previous paragraph. In Bi₂Se₃ surface, skew scattering is the main origin of the nonlinear transverse response since Berry curvature dipole is absent in trigonal symmetric systems[25]. The magnitude of $\frac{|E_y^{(2)}|}{(E_x^{(1)})^2}$ in the few-layer WTe₂, TaIrTe₄, and Bi₂Se₃ surface was $\sim 10^{-4}$–$10^{-2}$ μmV⁻¹. In our PbTaSe₂ samples, the conductivity is significantly larger than that in other materials, and $\frac{|E_y^{(2)}|}{(E_x^{(1)})^2}$ also shows large values even in the normal state ($\frac{|E_y^{(2)}|}{(E_x^{(1)})^2}$ approximately $10^2$ μmV⁻¹). Interestingly, it appeared that the data of the few-layer WTe₂, bulk TaIrTe₄, Bi₂Se₃ surface, and present PbTaSe₂ were aligned in one line in this plot, potentially revealing the universal feature of the scattering-induced nonlinear transverse response. Moreover, the $\frac{|E_y^{(2)}|}{(E_x^{(1)})^2}$ values became even larger by two orders of magnitude in the SC state. Although we cannot simply compare the nonlinear transverse signals in the normal state and

those in the SC state, we can clearly acknowledge the remarkable enhancement of the nonlinear anomalous transport in the SC region in Fig. 4a, b. These results imply that the large conductivity in the normal state and the vortex dynamics in the SC state may be advantageous for giant anomalous nonlinear transport.

In summary, we studied the second-order nonlinear transport in trigonal superconductor PbTaSe₂ under the time-reversal symmetry condition. The observed nonlinear transverse response and intrinsic rectification effect satisfy the characteristic directional dependence of the trigonal symmetry. Furthermore, both signals are significantly enhanced around the superconducting transition, where the excitation of vortex/antivortex string pairs governs the resistance. The asymmetric vortex Hall effect is a plausible scenario for the observed nonlinear transport. The present results elucidate a new aspect of vortex dynamics in superconductors and pave the way for investigating new properties and functionalities in noncentrosymmetric conductors.

## Methods

**Device fabrication.** Bulk PbTaSe₂ single crystals were grown using a flux method in an evacuated quartz tube. Stoichiometric amounts of Pb, Ta, and Se were sealed in an evacuated quartz tube, and 50 mol% KCl and 50 mol% PbCl₂ were mixed. The quartz tube was heated at 900 °C for 24 h and then cooled to room temperature. After crystal growth, the flux was removed by dissolution in water. The obtained PbTaSe₂ single crystals were exfoliated into thin flakes using the Scotch-tape method, and the flakes were transferred onto a Si/SiO₂ substrate. The thickness of the exfoliated flakes was measured using atomic force microscopy. A Hall bar configuration was fabricated on the flakes with Au (150 nm)/Ti (9 nm) electrodes. The pattern was fabricated using electron beam lithography, and the electrodes were deposited using an evaporator.

In fabricating the Hall bar configuration on the exfoliated flakes, we judged the crystal orientation from the straight edges of the flakes, which can be assumed to be in the zigzag direction. It is known that straight edges in exfoliated transition-metal dichalcogenides are identical to zigzag directions with high probability[32]. Although PbTaSe₂ has intercalated Pb layers in TaSe₂, we also adopted this criterion to determine the crystal orientation of PbTaSe₂. After the transport measurement of

sample 1, it was double checked by the STEM measurement, as discussed in the main text. From the results of the STEM measurement for sample 1, we conclude that the above method of determining the crystal orientation can also be applied to PbTaSe$_2$. Schematic images of PbTaSe$_2$ and 2H-NbSe$_2$ in the main text and Supplementary Notes are drawn by VESTA[47].

**Transport measurements.** The first and second harmonic resistances were measured using AC lock-in amplifiers (Stanford Research Systems Model SR830 DSP) with a frequency of 13 Hz in a quantum design physical property measurement system.

As discussed in previous studies [4–25], the voltage in the noncentrosymmetric system can be given as follows:

$$V = R^{(1)}I + R^{(2)}I^2, \tag{1}$$

where the first and second terms represent linear and second-order nonlinear transport, respectively. In this study, we focus mainly on $R^{(2)}$ under a time-reversal symmetric condition, that is, without a magnetic field.

When an AC bias current with a frequency of $\omega$ ($I = I_0 \sin \omega t$) is applied, it leads to

$$
\begin{aligned}
V &= R^{(1)}I_0 \sin \omega t + R^{(2)}I_0^2 \sin^2 \omega t \\
&= R^{(1)}I_0 \sin \omega t + \frac{1}{2}R^{(2)}I_0^2 \left\{ 1 + \sin\left(2\omega t - \frac{\pi}{2}\right) \right\}.
\end{aligned}
\tag{2}
$$

Therefore, by extracting the first and second harmonic resistances, we obtain

$$R^{\omega} \equiv \frac{V^{\omega}}{I_0} = R^{(1)} \tag{3}$$

and

$$R^{2\omega} \equiv \frac{V^{2\omega}}{I_0} = \frac{1}{2}R^{(2)}I_0. \tag{4}$$

Next, we derive the expression for the normalized nonlinear transverse signal $\frac{E_y^{(2)}}{(E_x^{(1)})^2}$, where $E_y^{(2)}$ and $E_x^{(1)}$ are the second-order nonlinear electric fields in the transverse direction and the linear electric field in the longitudinal direction, respectively, when current is applied along the zigzag direction. $E_y^{(2)}$ is written as

$$E_y^{(2)} = \rho_{yx}^{(2)}j_x^2, \tag{5}$$

where $j_x$ is the current density and $\rho_{yx}^{(2)}$ is the second-order resistivity. By considering $V_y^{(2)} = WE_y^{(2)}$ and $I_x = Wtj_x$, where $V_y^{(2)}$ is the nonlinear transverse voltage, $W$ is the channel width, $t$ is the thickness of the flake, and $I_x$ is the current, it transforms into

$$V_y^{(2)} = \frac{\rho_{yx}^{(2)}}{Wt^2}I_x^2 = R_{yx}^{(2)}I_x^2 \tag{6}$$

Therefore, using $E_x^{(1)} = \rho_{xx}^{(1)}j_x$, where $\rho_{xx}^{(1)} = \frac{Wt}{L}R_{xx}^{(1)}$ is the linear longitudinal resistivity with channel length $L$, $\frac{E_y^{(2)}}{(E_x^{(1)})^2}$ is calculated as

$$\frac{E_y^{(2)}}{(E_x^{(1)})^2} = \frac{\rho_{yx}^{(2)}}{(\rho_{xx}^{(1)})^2} = \frac{L^2}{W}\frac{R_{yx}^{(2)}}{(R_{xx}^{(1)})^2} = \frac{2L^2}{WI_0}\frac{R_{yx}^{2\omega}}{(R_{xx}^{\omega})^2}. \tag{7}$$

**Selection rules for nonlinear transport under time-reversal symmetric condition in trigonal systems.** Nonlinear current density $\boldsymbol{j}^{(2)}$ in the noncentrosymmetric system is generally written as $\boldsymbol{j}^{(2)} = \beta \boldsymbol{EE}$ or $j_i^{(2)} = \beta_{ijk}E_jE_k$, where $\beta$ is a third-order tensor[48]. Considering PbTaSe$_2$ with point group D$_{3h}$, $\beta$ leads to

$$\beta = \begin{pmatrix} \beta_{11} & -\beta_{11} & 0 & 0 & 0 & 0 \\ 0 & 0 & 0 & 0 & 0 & -\beta_{11} \\ 0 & 0 & 0 & 0 & 0 & 0 \end{pmatrix} \tag{8}$$

Here, $x$, $y$, and $z$ are parallel to the armchair direction, parallel to the zigzag direction, and perpendicular to the plane, respectively. Therefore, $\boldsymbol{j}^{(2)}$ under electric field $\boldsymbol{E}$ is written as follows:

$$\boldsymbol{j}^{(2)} = \begin{pmatrix} \beta_{11}(E_x^2 - E_y^2) \\ -2\beta_{11}E_xE_y \\ 0 \end{pmatrix} \tag{9}$$

When the electric field is applied along the armchair direction ($E_x = E, E_y = 0$), $\boldsymbol{j}^{(2)}$ leads to

$$\boldsymbol{j}^{(2)} = \begin{pmatrix} \beta_{11}E^2 \\ 0 \\ 0 \end{pmatrix} \tag{10}$$

On the other hand, for the electric field applied along the zigzag direction ($E_x = 0, E_y = E$), $\boldsymbol{j}^{(2)}$ leads to

$$\boldsymbol{j}^{(2)} = \begin{pmatrix} -\beta_{11}E^2 \\ 0 \\ 0 \end{pmatrix} \tag{11}$$

In both cases, $\boldsymbol{j}^{(2)}$ has only the $x$-component. This directional dependence (selection rule) in trigonal systems was confirmed in this study.

## Data availability
The data that support the findings of this study are available from the corresponding author upon reasonable request.

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

## Acknowledgements

We thank T. Nojima, S. Koshikawa, S. Okazaki, H. Isobe, and N. Nagaosa for fruitful discussions. Y.M.I. was supported by the Advanced Leading Graduate Course for Photon Science (ALPS). T.I. was supported by JSPS KAKENHI grant numbers JP19H01819, grant from Yazaki Memorial Foundation for Science and Technology, JST PRESTO (grant no. JPMJPR19L1). T.S. was supported by JST CREST (grant no. JPMJCR16F2). This work was supported by JSPS KAKENHI grant number JP19H05602 and the A3 Foresight Program.

## Author contributions

Y.M.I., T.I., and Y.I. conceived the research project. H.N., and T.S. synthesized the bulk material. Y.M.I., and C.G. fabricated the microdevices, performed the experiments, and analyzed the data. S.H. performed the theoretical calculations. Y.M.I., T.I., S.H., and Y.I. wrote the manuscript. All authors have led the physical discussions.

## Competing interests

The authors declare no competing interests.
