## [Peer Review File · Nature Communications]

Giant second harmonic transport under time reversal symmetry in a trigonal superconductorThe previous round of reviews were completed at another journal

Responses to Reviewers' comments

Reviewer #1:

The manuscript “Giant intrinsic rectification and nonlinear Hall effect under time-reversal symmetry in a trigonal superconductor” described a systematic experimental study on the nonlinear transport properties of PbTaSe₂ crystals discovered both at its normal state and at the superconducting state. The authors attributed the strong normal state nonlinear Hall effect and longitudinal second harmonic signal to skew and side-jump scatterings.

Nonlinearity is inherent in the superconductor to normal metal phase transition, because the I-V curves are simply highly nonlinear at the critical current or at the critical temperature. Thus, it is not surprising there will be a second harmonic longitudinal resistance appears at these two transition points (R_{xx} data in figure 3a,3b, 3c & 3d).

Nonlinearity in non-central symmetric system is also not surprising (R_{xx} data in figure 2b & 2c). The interesting part of the manuscript is the discovery of the anomaly increase of the nonlinear Hall component at the superconducting phase transition temperature, and provided a plausible theoretical explanation. However, the current finding does not appear to be a significant enough breakthrough in the field of superconductivity that warrants publication in Nature Physics, and may be more suited for a more specialized journal. Here are the specific comments that need to be addressed in this manuscript:

Response

We would like to emphasize that the second harmonic response, which we observed in this work, cannot be attributed to the nonlinearity in the *I-V* characteristics of superconductors but an emergent novel transport reflecting the noncentrosymmetric crystal structure. Especially, as reviewer #1 pointed out, discovery of the nonlinear anomalous Hall effect in the superconducting state (emergence of the second-order nonlinear voltage only in the transverse direction) is an important significance of the present work.

Below we would like to explain it in detail.

First, the nonlinearity inherent to the superconducting transition (which reviewer #1 pointed out) generally cause the odd-order nonlinear responses, which is clearly distinct from the second harmonic response (even-order effect) studied in this work. In fact, as reviewer #2 mentioned in comment 6, third or fifth (odd-order) harmonic voltage has been commonly studied in superconductors (J. G. Ossandon *et al.*, *JPCS* **43**, 655 (2006)., for example). However, the second-order nonlinear transport, which indicates the rectification effect reflecting the symmetry breaking and thus a unique functionality of noncentrosymmetric crystals, cannot be explained by such a conventional effect. Actually, we confirmed that the second-harmonic voltage is absent in a centrosymmetric superconductor 2H-NbSe₂, which has

a same monolayer structure as PbTaSe_2 but a different stacking sequence (Response to comment 5 of reviewer #1).

The intrinsic rectification effect (second-order longitudinal nonlinear response or nonreciprocal transport) is now recognized as a sophisticated probe of symmetry breaking and related characteristics such as electron band deformations, asymmetric quasiparticles dynamics, and fluctuation effects in quantum phases (*Nat. Commun.* **9**, 3740 (2018)., *Nat. Mater.* <https://doi.org/10.1038/s41563-021-00992-7> (2021)). Recently, nonreciprocal transport under the magnetic field is extensively studied in a variety of noncentrosymmetric crystals including superconductors (*Nature* **584**, 373 (2020) etc.). By contrast, the nonreciprocal transport under time reversal symmetric condition is a highly nontrivial issue because the time-reversal symmetry operation connects two states with crystal momenta k and $-k$, thus leading to the same group velocity of the two states. For their mechanisms, effects of electron correlations and roles of the dissipation by scattering are discussed (*Sci. Rep.* **8**, 2973 (2018)). In the present work, we firstly report the nonreciprocal transport under the time reversal symmetric conditions.

Second, the second-order nonlinear transport along the transverse direction. i.e., the nonlinear anomalous Hall effect, is also a hot topic today. This nonlinear anomalous Hall effect is not only a new type of Hall effect but also an emergent probe of the band topology/geometry and the anomalous scattering process (*Phys. Rev. Lett.* **115**, 216806 (2015), *Nature* **565**, 337–342 (2019), *Nat. Mater.* **18**, 324–328 (2019), and *Nature Review Physics* (2021) DOI: 10.1038/s42254-021-00359-6). So far, it has been reported in a few compounds with low symmetry (space group with only one mirror plane) but, in principle, the nonlinear anomalous Hall effect with distinctive origins can occur in crystals with higher symmetry. For example, the nonlinear Hall effect by the skew scattering of chiral Bloch electrons in trigonal systems is theoretically discussed (*Sci. Adv.* **6**, eaay2497 (2020)). Search for the materials which show the nonlinear anomalous Hall effect with such a new origin is an important subject. Moreover, exploration of the nonlinear anomalous Hall effect in exotic quantum states, such as superconductivity, is a significant challenge from both fundamental and technological points of view. In this paper, we report the nonlinear anomalous Hall effect in trigonal crystals and its anomalous enhancement in the superconducting state for the first time, revealing the new perspective of the noncentrosymmetric superconductors.

We successfully explained both the longitudinal and transverse second-order nonlinear transport mentioned above by considering the anomalous scattering process (in the normal state) and the vortex/anti-vortex dynamics (in the superconducting state).

On this basis, we believe that our findings would attract broad attention from audience working on superconductivity, topological material science, low dimensional materials, and any other symmetry-related physics.

Comment 1

1. First of all, the key findings in the manuscript seem to be taken with a relatively large background noise. 375nV shouldn't be a signal that is very difficult to measure (R_{xy} signal in figure 3a). The authors need to improve the signal to noise ratio.

Response

We detect the nonlinear resistance superposed to the linear resistance by the lock-in technique. Because the nonlinear component is smaller than the linear component by two orders of magnitude, it is difficult to detect the nonlinear signals especially for the low current region. Therefore, the noisy background is inevitable. However, we can average the several scans to statistically reduce the background noise. We remeasured the current dependence of second harmonic resistance and averaged them together with the original data (Figs. R1a and c, previous Figs. 3a and b in the main text). The result is shown in Figs. R1b and d (present Figs. 3a and b in the main text). The signal-to-noise ratio is improved compared to the previous data. We replaced the previous figures to the present ones in the main text.

Figure R1. Improvement in the signal-to-noise ratio. a-d, Current dependence of $R^{2\omega}$ (left) and R_{xx}^{ω} (right) at $T = 2$ K in (a,b) configuration A and (c,d) configuration B. Red and blue lines indicate longitudinal ($R_{xx}^{2\omega}$) and transverse ($R_{yx}^{2\omega}$) resistance, respectively. Figures a and c

show the original data, which is identical to previous Figs. 3a and b in the main text. Figures b and d show the present data after reducing the noise statistically.

Comment 2

2. The nonlinear effects in the normal state seem to be taken at a completely different current regime as compares to the superconducting states. The authors should explain why the data stops at 0.5mA (figure 2b). From the data, it seems that R_{2w} will become negative at current below 0.5mA, if the linear trend continuous.

Response

We focused on the second-order nonlinear voltage in the relatively large current region since the signal in the low current region seems to be too small to observe. (Note that the magnitude of the second-order nonlinear transport linearly increases by the applied current.)

When we measure the linear or nonlinear resistance by using a lock-in amplifier, we obtain two pieces of information about the electrical response, i.e., the absolute value of voltage R and the phase of the response θ . Here, the first (second) harmonic voltage R^ω ($R^{2\omega}$) is obtained as $R^\omega = R \cos \theta / I$ ($R^{2\omega} = R \sin \theta / I$), where I is the applied current. When we successfully measure the first (second) harmonic voltage, θ should be around 0° ($\pm 90^\circ$). In Figs. R2a and b, we show the second harmonic transverse resistance $R_{yx}^{2\omega}$ and θ , respectively. In the high current region above 1 mA, θ is almost 90° , indicating the reliability of the second harmonic signals. Between 1 mA and 0.5 mA, θ deviates from 90° and both $R_{yx}^{2\omega}$ and θ become noisy below 0.5 mA. This is because the nonlinear signals decrease with decreasing the current and the signal becomes too small to detect properly. $R_{yx}^{2\omega}$ seems to be slightly negative around 0.25 mA but θ is not around -90° so that the data are not reliable.

Figure R2. Transverse second harmonic resistance and its phase. a,b, Current dependence of (a) $R_{yx}^{2\omega}$ and (b) θ at $T = 20$ K. Below 0.5 mA, both $R_{yx}^{2\omega}$ and θ become noisy.

Comment 3

3. Theoretically, the motion of vortex-anti-vortex does not rely on crystal directions, but the anomalous nonlinear Hall signal is very different from the manuscript. The authors will have a detail discussion on this issue.

Response

In the linear response regime, the motion of vortices is theoretically independent of its directions. In contrast, the higher-order nonlinear transport generally includes the information of the underlying lattice symmetry. This is an advantage of the nonlinear transport study. In fact, a theoretical model we discussed in the present work (Eq. S5 in Supplementary Materials) and observed directional dependence of the second-order nonlinear transport (Figs. 2 and 3) is fully consistent with the trigonal symmetry.

Comment 4

4. The largest signal is taken at the superconductor-normal metal phase transition, with a relatively large current (0.1mA to 0.8mA), which could cause significant heating effect. The authors should provide evidence that the measured data is not influenced by heating.

Response

We would thank reviewer #1 for this important comment. We found that the heating effect does not affect the second harmonic resistance as discussed below.

We consider the heating effect on the transport coefficients caused by the external current in the similar manner to the previous studies (T. M. Mishonov *et al.*, *Eur. Phys. J. B* **26**, 291 (2002) and J. G. Ossandon *et al.*, *JPCS* **43**, 655 (2006).). The current-voltage relation is in general written as

$$V = R^{(1)}(T)I + R^{(2)}(T)I^2 + R^{(3)}(T)I^3 + \dots$$

where T is the temperature of the sample. The Joule heating is accounted by the power $P = VI$. The energy transfer from the sample to the environment with the temperature T_0 is given by $P = G\delta T$, where G is the thermal boundary conductance and $\delta T = T - T_0$ represents the temperature variation. Then, the temperature change δT is expressed as a function of the current, and the leading-order term is the square of the current. We thereby obtain

$$V = R^{(1)}(T_0)I + R^{(2)}(T_0)I^2 + \left[R^{(3)}(T_0) + \frac{R^{(1)}(T_0)R^{(1)'}(T_0)}{G} \right] I^3 + \dots$$

where $R^{(1)'} = \partial_T R^{(1)}$ is the temperature derivative. It is notable that there is a qualitative difference between the second-order and third-order terms. Namely, the third-order contribution can enter through the combination of the first-order resistance and its temperature-varying effect, but the second-order coefficient is not generated by such an effect. Hence the second-order signals studied in this work captures only the intrinsic effect from the nonreciprocal response in the noncentrosymmetric superconductors.

We also note that the directional dependence of the second harmonic signals and its reproducibility observed in plenty of different samples (discussed in the main text and Supplementary Materials) cannot be explained by the heating effect but can be purely attributed to the effect of the crystal symmetry.

We added the above discussion in Supplementary Materials.

Comment 5

5. As a comparison, similar data should be acquired from superconductor with inversion symmetry as supplement to the manuscript

Response

We thank Reviewer #1 for this important suggestion. As a comparable centrosymmetric superconductor, we investigated the second-order nonlinear response in 2H-NbSe₂. In PbTaSe₂ (Fig. R3a), all the TaSe₂ layers are stacking in the same direction, leading to the noncentrosymmetric crystal structure (space group of $P\bar{6}/m2$) as shown in the main text. With contrast, in 2H-NbSe₂ (Fig. R3b), neighboring NbSe₂ layers are rotated by 180°. Thus, bulk 2H-NbSe₂ becomes centrosymmetric (space group of $P6_3/mmc$). We prepared the 2H-NbSe₂ device (thickness $t = 103$ nm) in configuration A (current flowing along the zigzag direction as shown in Fig. R3c) and measured the first and second harmonic resistance similarly to PbTaSe₂ (Fig. 3a in the main text). 2H-NbSe₂ shows metallic behavior and superconductivity around $T = 7$ K (Fig. R3d).

First, we focus on the nonlinear signals in the normal state. Figure R4a (b) shows $R_{xx}^{2\omega}$ and $R_{yx}^{2\omega}$ as a function of the current in PbTaSe₂ (2H-NbSe₂) at $T = 20$ K. PbTaSe₂ shows the large second harmonic signal in $R_{yx}^{2\omega}$ while $R_{xx}^{2\omega}$ is small (nonlinear anomalous Hall effect). With contrast, in 2H-NbSe₂, both $R_{xx}^{2\omega}$ and $R_{yx}^{2\omega}$ are indiscernible. This indicates that second harmonic signal is truly absent in a centrosymmetric system.

Next, we want to discuss the nonlinear signals in the superconducting state. Figure R4c (d) shows the current dependence of $R_{xx}^{2\omega}$ and $R_{yx}^{2\omega}$ in PbTaSe₂ (2H-NbSe₂) at $T = 2$ K. $R_{yx}^{2\omega}$ shows the peak structure during the superconducting transition in $R_{yx}^{2\omega}$ while the indiscernible signal in $R_{xx}^{2\omega}$ as discussed in the main text. With contrast, both $R_{xx}^{2\omega}$ and $R_{yx}^{2\omega}$ in 2H-NbSe₂ are negligibly small, which is the expected behavior of the centrosymmetric crystals. We note a small peak structure in $R_{xx}^{2\omega}$ at the transition, which also appears in $R_{xx}^{2\omega}$ of PbTaSe₂. This might come from the asymmetry in electrical contacts or inhomogeneity of the superconductivity. However, its magnitude is much smaller than the intrinsic signals of the nonlinear anomalous Hall effect in PbTaSe₂ (Fig. R4c).

We added this argument in Supplementary Materials.

Figure R3. Schematic crystal structures of noncentrosymmetric PbTaSe₂ and centrosymmetric 2H-NbSe₂, and superconducting property in 2H-NbSe₂. **a,b**, Schematic crystal structures of **(a)** PbTaSe₂ and **(b)** 2H-NbSe₂. In PbTaSe₂, all TaSe₂ layers are stacking in the same direction, leading to the noncentrosymmetric crystal structure. On the other hand, in 2H-NbSe₂, neighboring NbSe₂ layers are rotated by 180°, resulting in the centrosymmetric crystal structure. Red wedges indicate the direction of each TaSe₂/NbSe₂ layers. **c**, Schematic image of the applied current along zigzag direction in TaSe₂/NbSe₂ layer. **d**, Temperature dependence of R_{xx}^{ω} in 2H-NbSe₂ when $I = 50 \mu\text{A}$. The inset shows R_{xx}^{ω} around the superconducting transition ($T_c = 7$ K).

Figure R4. Comparison between noncentrosymmetric PbTaSe₂ and centrosymmetric 2H-NbSe₂. **a,b**, Current dependences of the second harmonic resistance $R^{2\omega}$ in **(c)** PbTaSe₂ and **(d)** 2H-NbSe₂ at $T = 20$ K. **c,d**, Current dependence of $R^{2\omega}$ (left) and R_{xx}^ω (right) at $T = 2$ K in **(e)** PbTaSe₂ and **(f)** 2H-NbSe₂. In Figs. a-d, red and blue lines indicate longitudinal ($R_{xx}^{2\omega}$) and transverse ($R_{yx}^{2\omega}$) resistance, respectively.

Comment 6

6. The first paragraph is an important portion of the manuscript for the readers to understand the topic, and the authors should provide appropriate citations to the literature. Currently there is no citation in the first paragraph of the manuscript.

Response

Because the abstract does not need references in Article, we have no reference in the abstract. Instead, we added several references in the first paragraph.

Reviewer #2:

The manuscript reports an observation of the second harmonic voltage peaks near the critical temperature in the trigonal superconductor with broken inversion symmetry. The effect is interpreted in terms of the rectified motion of the spontaneous vortex-antivortex pairs.

I find the experimental part very interesting and potentially suitable for publication in *Nature Physics*. The reason is that usually superconductors have the third harmonic nonlinearity while the present material shows the second harmonic one. However, the theoretical interpretation written in the supplementary material (SM) generates several important questions.

Response

We would appreciate reviewer #2 for her/his recognition of the importance of our study and her/his recommendation for publication in *Nature Physics*. Below, we have addressed the reviewer's comments with appropriate changes in the manuscript.

Comment 1

1. First, the suggested model assumes that the density of spontaneously created vortices is rather large, of the order of $1/\xi^2$ (line 218 of SM). This density seems to be independent of the current. This means that the superconductor is always close to the transition to the normal state, because at such vortex density their cores almost overlap. This assumption can (and should) be tested by some other measurements, e.g. thermal conductivity or tunneling spectroscopy.

Response

In general, the number of vortices should depend on the current. For the low-current limit, the formula based on the vortex unbinding is known as in Eq. (S16) in Supplementary Materials. However, if we estimate the number of vortices based on this expression, it exceeds the maximum and therefore we need another expression. Then, in our model, we introduce the number density $n_v = \frac{\alpha'}{\pi\xi^2}$ phenomenologically as discussed in Supplementary Materials Section IV-1, in order to make an order estimation of the signal theoretically. We also emphasize that estimated values of the ratio of the nonlinear anomalous Hall resistance to the linear resistance (Fig. S4c) is not dependent on the number of vortices, which is of the same order as the ones in experiments.

Experimentally, it is not easy to detect the number of vortices and/or vortex profiles due to the noise of the signals under the finite current where the system is in a non-equilibrium state. As for the current dependence of the signal, we newly added the experimental data under the different currents of $I = 60 \mu\text{A}$ and $I = 100 \mu\text{A}$ (Fig. R5), as will be also answered in the

Comment 3 of reviewer #2 below. The magnitudes for these two cases are not very different, indicating that the current dependence is not strong in such a current regime and our current-independent vortex density should be valid in a qualitative level, although it is not quantitatively.

Figure R5. Temperature dependence of transverse second harmonic resistance at different current. Blue circles and green triangles show $R_{yx}^{2\omega}$ at $I = 60 \mu\text{A}$ and $100 \mu\text{A}$, respectively.

Comment 2

- (a) 2. In order to obtain the asymmetric Hall effect for vortices there is a ratchet potential introduced directly into the model. The origin of this potential is not clear. It is mentioned that it comes from the pinning centers but why all pinning centers should have the identical anisotropy?
- (b) The application of the Fokker-Plank theory to the vortex motion in the pinning potential is also very questionable. I did not see such approaches in the literature.
- (c) But the most striking thing is the final result Eq. S12 or S15. The effect grows with l_v , that is with decreasing the density of the pinning centers which provide the ratchet effect! I find this counter-intuitive and probably wrong.

Response

- (a) The microscopic justification for the pinning potential is actually difficult: it is dependent on the lattice structures, defects, impurities, and these objects can all be pinning centers. The strong one itself can pin the vortices, while the weak ones may pin the vortices in a collective manner (collective pinning). Since these microscopic details are not available, we phenomenologically introduce the simple asymmetric potential in the noncentrosymmetric systems and describe the motion of vortices based on the Langevin equation, which qualitatively reproduce the magnitudes of signals observed in

experiments.

We have assumed the completely periodic structure and neglected the randomness in the vortex potential. We expect that such randomness is not relevant for the nonreciprocal signals since the asymmetry of the potential and the pinning effects are essential for the vortex transports in the noncentrosymmetric systems and are accounted without the randomness.

In the revised text, we have added more information on the origin of vortex potentials in Sec. IV of Supplementary Materials.

- (b) The vortex dynamics in pinning potentials is sometimes treated as a Brownian motion of the point particle as discussed in, e.g., B.-Y. Zhu *et al.*, *Phys. Rev. B* **68**, 014514 (2003) and V. Ambegaokar *et al.*, *Phys. Rev. B* **21**, 1806 (1980), and vortex motions are described by solving the Fokker-Planck equation. We have also utilized a similar theoretical framework to evaluate the vortex motions in Ref. 15 in the Supplementary Materials (S. Hoshino *et al.*, *Phys. Rev. B* **98**, 054510 (2018)). To clarify the validity of the Fokker-Planck theory in the present case, we added the detailed derivation and explanation of the Fokker-Planck equation in Sec. IV of Supplementary Materials.
- (c) We understand the Reviewer #2's concern that eq. S15 is seemingly counter-intuitive. As Reviewer #2 pointed out, the ratchet effect should become weaker if the size of the pinning potential is a constant and only ℓ_v increases. However, it should be noted that the size of the pinning potential ($c\ell_v$) will also expand with increasing the periodicity of the pinning potential ℓ_v (Fig. R6) in the present model. This is because we fixed c (the ratio of the pinning potential size to ℓ_v) in this model. Thus, we naturally expect the larger nonlinear transport signals with the increase of ℓ_v . This assumption is justified by the following discussion.

As mentioned in the above reply (b), we treated vortex motions as a Brownian motion of the point particle. In reality, however, vortex has a finite size, which depends on the temperature. Therefore, we incorporated the temperature-dependent variation of the vortex size into ℓ_v and $c\ell_v$. In other words, the reduction of the vortex size with decreasing the temperature can be effectively simulated by the increase of ℓ_v .

Specifically, we deduced the relation $\ell_v \propto \sqrt{T_c - T}$ to express the above situation. This relation can be confirmed in Supplementary Materials as follows. In Fig. S4b, we first checked that the critical current I_{pin} behaves as $I_{\text{pin}} \propto \sqrt{T_c - T}$. In addition to this result, by using the two general expressions of the pinning potential $U \propto c\ell_v I_{\text{pin}}$ (which was evaluated from the depinning process of vortices) and $U \propto T_c - T$ (which was estimated from the energy gain of vortices), we can conclude that the relation $\ell_v \propto (T_c - T)/I_{\text{pin}} \propto \sqrt{T_c - T}$ actually holds in the present case. Thus, the obtained theoretical model well reproduces the experimental result of the temperature dependence of the second harmonic signal (Supplementary Materials Fig. S4d).

In addition, we can also explain the presence of ℓ_v in q_2 from the dimensional analysis. In our model, we have the four dimensionful parameters η_0 [kg/s], ℓ_v [m], U_0 [J], $k_B T$ [J]. In order to have the dimension of q_2 [$\text{kg}^{-2}\text{m}^{-1}\text{s}^3$], the length ℓ_v must be multiplied for q_2 only once, to result in the expression in our paper. More explicitly, we assume the form $q_2 = (\text{dimensionless part}) \times \eta_0^x \ell_v^y U_0^z (k_B T)^w$ and compare the dimensions in the left- and right-hand sides, to result in $x = -1, y = 1, z + w = -1$. The result is also consistent with the previous study (Ref. 15 in Supplementary Materials, S. Hoshino *et al.*, *Phys. Rev. B* **98**, 054510 (2018).) where the periodicity appears in the nonreciprocal transport signal. We have explained these points in Sec. IV of Supplementary Materials.

Figure R6. Schematic image of asymmetric pinning potentials. With the increase of ℓ_v , the size of the pinning potential expands simultaneously.

Comment 3

- (a) 3. The additional important parameter in the model is the pinning force which temperature dependence is suggested to be the reason of the peaked temperature dependence of the second harmonic signal. Namely, it is written (line 223 SM) “The absence of the signal in low-temperature region is not due to the vortex-antivortex binding but to the pinned vortices.” According to this logic, the low-temperature signal should grow with increasing current density. That is, for larger currents the peak of $R_2 \propto \omega$ should transform into the broader feature. It would be useful to see the experimental confirmation of this behavior. At the present version only one value of the current for each configuration is considered (Fig. 3c,d).
- (b) By the way, for some reason the considered currents are strongly different (0.06 mA and 0.3 mA). What is the reason for such a difference?

Response

- (a) As reviewer #2 pointed out, the signal at low temperatures grows with increasing current. Figure R7 shows $R_{yx}^{2\omega}$ at $I = 60 \mu\text{A}$ (blue circles) and $100 \mu\text{A}$ (green triangles). With increasing the current, the peak structure in $R_{yx}^{2\omega}$ becomes broad, indicating pinned vortices at the low temperature region.
- (b) The different values of the current come from the thickness difference of the device and also from the sample variation of the critical current density. Considering the device geometries (sample thickness and electrodes width) of samples 1 and 2, the current densities are calculated as $3.3 \times 10^8 \text{ A/m}^2$ and $6.8 \times 10^8 \text{ A/m}^2$, respectively, whose magnitudes are in the same order. In this work, we focused on the temperature dependence of the second harmonic resistance at the current value close to the critical current. In samples 1 and 2, the critical currents defined as the midpoint of the resistive transition at $T = 2 \text{ K}$ are 0.14 mA and 0.52 mA , respectively, which are also slightly different and contribute to the different current values in the measurement.

Figure R7. Temperature dependence of transverse second harmonic resistance at different current. Blue circles and green triangles show $R_{yx}^{2\omega}$ at $I = 60 \mu\text{A}$ and $100 \mu\text{A}$, respectively.

Comment 4

4. In general, the model of spontaneous vortex pairs is moved totally in the SM. But since it is quite an important part of the work, I think it would be useful to present some key assumptions and estimations in the main text.

Response

We added the key assumptions and the estimations in the main text.

Comment 5

Finally, there are some comments about the presentation.

(a) Line 460 main text, it is written that Fig. 4b shows the function of temperature but in fact the horizontal axis name is $(\sigma_{xx}^{\omega})^2$

(b) Line 434 main text, it is written that “Directional dependence of the nonlinear transport illustrated in Fig. 1(a) is confirmed.” But Fig. 1(a) shows just the lattice and no information about the transport.

Response

Thank you for pointing out our mistakes. We revised the figure captions.

Comment 6

To conclude, I encourage authors to improve the theoretical model in order to make the story more convincing. Can one think of other possible explanations besides the spontaneous vortices? For example, can the second harmonic peak have a similar origin as the usual third harmonic peak (<https://iopscience.iop.org/article/10.1088/1742-6596/43/1/160/meta>) modified by the lack of the inversion symmetry?

Response

We would thank for informing us this important previous research. This work discusses the mechanism due to the temperature oscillation caused by the applied ac current. This possibility is ruled out in our case because the heating effect is not relevant for the second-harmonic signal. Notably, such an effect enters only for the third or higher-order contribution. Hence our mechanism is different from the heating effect discussed in the previous work.

We added the above discussion in the Sec. VIII of Supplementary Materials and cited the above literature.

As for the other possible mechanisms, in addition to the vortex dynamics we considered in this manuscript, there might be also the amplitude fluctuation effect when the system is close to the transition temperature. The paraconductivity for the second-harmonic response may contribute to the behavior near the mean-field transition temperature, although at lower temperature the phase fluctuation from vortex dynamics becomes dominant.

In the main text, we have included this comment on the possible other mechanisms.

Reviewer #3:

The manuscript by Itahashi et al. reported nonlinear electrical responses in PbTaSe₂, a trigonal, non-centrosymmetric superconductor. Recently, studying nonlinear responses in novel quantum materials is a hot topic. In particular, the present work focuses on the nonlinear responses in the electrical transport regime and in the superconducting states. Both aspects are highly interesting. The main findings of the paper include (1) observation of both longitudinal and transverse nonlinear electrical transport responses in the presence of time-reversal symmetry and (2) an enhancement of the response near the superconducting phase transition. Given the intriguing topic and the interesting findings, the paper is suitable for *Nature physics*. However, before I can make a decision, the following crucial technical points need to be carefully addressed.

Response

We would appreciate reviewer #3 for her/his recognition of the significance of our study and her/his recommendation for publication in *Nature Physics*. Below, we have addressed the reviewer's comments with appropriate changes in the manuscript.

Comment 1

1. The claim of observation of the nonlinear Hall effect is not quite correct. The authors need to carefully and thoroughly address this point. Yes, the authors have observed transverse nonlinear responses. However, this transverse response is NOT a Hall effect. There are two ways to appreciate this point.

First, we can do a simple symmetry analysis of the group D_{3h} (PbTaSe₂). We find that there is only one independent tensor component. Specifically, we have $xxx = -xyy = -yxy = -yyx$. The focus on the present paper is xxx and xyy . Yes, xxx is purely longitudinal and xyy is purely transversal. However, we see that they are NOT independent at all because $xxx = -xyy$. In fact, they are basically the same effect, i.e., second-order current rectification in the second order.

Second, for the second-order effect to be a Hall effect, the current-carrying state needs to host a nonzero total Berry curvature (or similarly a magnetization). In other words, the system would need to be able to host the kinetic magneto-electric effect (also known as the Edelstein effect). The group of D_{3h} does not support a current-induced magnetization. Therefore, a second-order Hall effect is impossible.

Essentially, what the authors observed are second-order current rectification in the second order. The claim of the nonlinear Hall effect is wrong. Rectification can be both longitudinal and transversal.

Although nonlinear Hall effect is not possible, I think the observation of intrinsic rectification without B-field and the enhancement near superconducting phase transition are already very interesting

Response

We are afraid that there might be a slight confusion about the definition of the nonlinear anomalous Hall effect.

Reviewer #3 thinks the nonlinear transverse response in the trigonal systems is not the nonlinear Hall effect because the tensor components of the second-order longitudinal and transverse responses are not independent. However, the second-order transverse response under the application of the longitudinal AC current (Fig. R8a) is generally called the nonlinear Hall effect, which can be an excellent probe of the exotic electronic structure, band topology/geometry, and charge dynamics as seen in several papers including *Nature Review Physics* (2021) DOI: 10.1038/s42254-021-00359-6 and *Nature Materials* (2021) DOI: 10.1038/s41563-021-00992-7. Although the nonlinear Hall effect originating from the Berry curvature dipole (and also the current induced magnetization) will be absent in the trigonal systems as Reviewer #3 pointed out, the nonlinear Hall effect by other mechanisms can occur (Fig. R8b). In fact, the nonlinear Hall effect reflecting the skew scattering of chiral Bloch electrons in trigonal systems is theoretically suggested (*Sci. Adv.* **6**, eaay2497 (2020)). What we report in the normal state of PbTaSe₂ in the present manuscript can be considered to be the first experimental observation of this phenomenon. More importantly, we found the large enhancement of the nonlinear Hall effect in the superconducting region, which we believe to be a discovery of the nonlinear Hall effect in exotic quantum phases, and enrich our understanding and scope of novel nonlinear transport phenomena.

We also would like to emphasize the significance of our discovery of the intrinsic rectification effect under the time reversal symmetry, as reviewer #3 pointed out.

The intrinsic rectification effect (second-order longitudinal nonlinear response or nonreciprocal transport) is now recognized as a sophisticated probe of symmetry breaking and related characteristics such as electron band deformations, asymmetric quasiparticles dynamics, and fluctuation effects in quantum phases (*Nat. Commun.* **9**, 3740 (2018)., *Nat. Mater.* <https://doi.org/10.1038/s41563-021-00992-7> (2021)). Recently, the nonreciprocal transport under the magnetic field is extensively studied in a variety of noncentrosymmetric crystals including superconductors (*Nature* **584**, 373(2020) etc.). By contrast, the nonreciprocal transport under time reversal symmetric conditions is a highly nontrivial issue because the time-reversal symmetry operation connects the two states with crystal momenta k and $-k$, thus leading to the same group velocity of the two states. Effects of electron correlations and role of

the dissipation by scattering are discussed (*Sci. Rep.* **8**, 2973 (2018)). In the present work, we firstly report the nonreciprocal transport under the time reversal symmetric conditions. In addition, we extend this effect to the superconducting state and found a new perspective of the noncentrosymmetric superconductors.

On this basis, we believe that our findings would attract broad attention from audience working on superconductivity, topological material science, low dimensional materials, and any other symmetry-related physics.

Figure R8. Schematics and summary of nonlinear anomalous Hall effect. **a**, Schematics of nonlinear anomalous Hall effect. When the nonlinear signals appear perpendicular to the applied current, this is called nonlinear anomalous Hall effect. **b**, Summary of experimental works on nonlinear anomalous Hall effect.

Comment 2

2. About the enhancement near the superconducting phase transition, the authors gave the interpretation based on the asymmetric Hall effect of vortex-antivortex string. I found that the evidence for this is rather weak. Maybe it is a viable microscopic picture, but I think right now the results cannot unambiguously demonstrate this. I think that the authors should adjust their presentation to make sure to express that this is only a tentative and possible explanation.

Response

As the reviewer #3 pointed out, there are several possible mechanisms for explaining the nonlinear signals. We have additionally considered the effect of Joule heating, which has been

considered for the third or higher- harmonic signals, as suggested by reviewer #2 in comment 6. We found that this effect does not contribute to the second-harmonic signal and generates only the odd-order higher-harmonics. The detailed discussion is given in Sec. VIII of Supplementary Materials.

As discussed in the main text, the vortex motion is one of the plausible scenarios of the second-order nonlinear transport, but there is also another potential effect of the superconducting fluctuations: the phase and amplitude fluctuations of the order parameters can also contribute to the transport coefficients when the system is close to the mean-field transition temperature.

We have commented on these possibilities in the revised main text.

List of Revisions

Main text

1. Line 159, page 7

We added

“Although we note another possible contribution from the amplitude fluctuations above the superconducting transition temperature, we mainly focus on the vortex/antivortex dynamics in this work because it will be dominant below the transition temperature and the following theoretical model can also semiquantitatively explains the results.”

2. Line 188, page 8

We added

“which are determined from the detailed shape of the asymmetric pinning potentials by using Fokker-Planck equation (Supplementary equation S9)”

3. Line 192, page 8

We revised from

“Employing the realistic values of the phenomenological parameters, we estimated the value of $R_{yx}^{2\omega}$ to be approximately 1.4 m Ω near the superconducting transition temperature (Supplementary Fig. S4b), which is in good agreement with the experimental results.”

to **“We employed the realistic phenomenological parameters: The potential U and the length ℓ_v are determined from the experimental current density and magnetic field where the vortices are depinned. The number of vortices is determined from the maximum value because nonlinear responses are generated by thermal or current-noise fluctuations near the transition point (Supplementary Materials Section IV). Thus, we estimated the value of $R_{yx}^{2\omega}$ to be approximately 1.4 m Ω near the superconducting transition temperature (Supplementary Fig. S4b), which is in good agreement with the experimental results.”**

4. Line 201, page 9

We added

“The detailed analysis on the vortex pinning profile and superconducting fluctuation effect will further clarify the nature of nonreciprocal signals.”

5. Line 251, page 11

We revised from

“responsible”

to **“a plausible scenario”**.

References

We added the following references.

1. Shen, Y. R. *The Principles of Nonlinear Optics*. (Wiley, 1984).
2. Sánchez, J. C. R. *et al.* Spin-to-charge conversion using Rashba coupling at the interface between non-magnetic materials. *Nat. Commun.* **4**, 2944 (2013).
3. Armitage, N. P., Mele, E. J. & Vishwanath, A. Weyl and Dirac semimetals in three-dimensional solids. *Rev. Mod. Phys.* **90**, 015001 (2018).

Figures

1. Line 462, page 21

We revised from

“Directional dependence (selection rule) of the nonlinear Hall/rectification effect in trigonal PbTaSe₂.”

to “**Schematics of crystal structure and expected directional dependence (selection rule) of the nonlinear Hall/rectification effect in trigonal PbTaSe₂.**”.

- 2.

We replaced Figures 3a and b to new figures.

3. Line 494, page 23

We revised from

“as a function of temperature”

to “**as a function of $(\sigma_{xx}^{\omega})^2$** ”.

Supplementary Materials

- 1.

We added Supplementary Materials Section VIII and IX.

2. Line 162, page 11

We added

“**In order to determine the response coefficients, we focus on the case with the external field along the y-direction ($F_x = 0, F_y \neq 0$). According to Eq. S5, only the motion in the y-direction is involved, and then we consider the one-dimensional kinetic equation:**

$$\eta_0 \partial_t X = -\partial_X U + F + \xi(t) \quad (\text{S8})$$

where X is the coordinate of the particle, ξ represents the force from a thermal noise which satisfies the random average $\langle \xi(t) \xi(t') \rangle = 2\eta_0 k_B T \delta(t - t')$. The form of the

potential is determined later. We define the positional distribution function $P(x, t) = \langle \delta(x - X(t)) \rangle$, which satisfies the Fokker-Planck equation

$$\eta_0 \partial_t P = \partial_x [(\partial_x U - F)P + k_B T \partial_x P] \quad (\text{S9})$$

(See, for example, Appendix A5 of Ref. 16 for a simple derivation). The solution of this equation can be found in Refs 15 and 17. At the stationary condition $\partial_t P = 0$, we obtain the velocity

$$v = \langle \partial_t X \rangle = \int dx P(x) \frac{1}{\eta_0} (-\partial_x U + F) \quad (\text{S10})$$

as a function of the force. q_1 and q_2 are then obtained as the coefficients of F and F^2 , respectively.”.

3. Line 187, page 13

We added

“The form of Eq. S11 itself can be deduced also from the dimensional analysis by recognizing that there exist the four dimensionful parameters η_0 [kg/s], ℓ_v [m], U_0 [J], $k_B T$ [J].”

4. Line 193, page 13

We added

“While we have introduced the simplified ratchet potential phenomenologically by regarding the vortex as a point particle, it is difficult to derive the potential from the microscopic point of view. It may originate from the impurities and lattice defects, which pin the vortex in a collective manner for the weak pinning and the vortex may be pinned by a single center for the strong pinning. Our purpose here is to confirm the validity of the ratchet vortex scenario and hence we deal with the effective models introduced above. We also note that the randomness of the potential is neglected for simplicity, since the pinning effect and asymmetric feature, which are needed for the nonreciprocal transport, are accounted by our model.”

5. Line 228, page 14

We revised from

“The potential height may also be estimated as the condensation energy gain”

to “In order to estimate the temperature dependence of potential height, we employ the condensation energy gain”

6. Line 230, page 14

We revised from

“In addition, we assume $\ell_v \propto \sqrt{T_c - T}$ which corresponds to the fact that the size of vortices increases as $T \rightarrow T_c$ and the potential periodicity for vortices becomes effectively shorter

together with the increasing coherence length ξ_{ab} . With these results we obtain $I_{\text{pin}} \propto \sqrt{T_c - T}$.”

to “**In addition, we assume the temperature dependence of ℓ_v as $\ell_v \propto \sqrt{T_c - T}$, which corresponds to the fact that the size of vortices increases as $T \rightarrow T_c$ and the potential periodicity for vortices becomes effectively shorter together with the increasing coherence length ξ_{ab} . That is to say, the temperature-dependent variation of the vortex size, which cannot be treated in our model of Brownian motion of the point particle, was incorporated into ℓ_v . By considering the two general expressions of the pinning potential $U \propto c\ell_v I_{\text{pin}}$ (which was evaluated from the depinning process of vortices) and $U \propto T_c - T$ (which was estimated from the energy gain of vortices) along with $\ell_v \propto \sqrt{T_c - T}$, we obtain $I_{\text{pin}} \propto \sqrt{T_c - T}$.”**

7. Line 261, page 16

We added

“**Here we use the current-independent expression for the number of vortices, which is motivated by the fact that the observed magnitudes at different currents does not strongly deviate as shown in Fig. S4e.**”

Supplementary references

We added the following references.

15. Ambegaokar, V., Halperin, B. I., Nelson, D. I. & Siggia, E. D. Dynamics of superfluid films. *Phys. Rev. B* **21**, 1806–1826 (1980).
16. Zhu, Y., Marchesoni, F., Moshchalkov, V. & Nori, F. Controllable step motors and rectifiers of magnetic flux quanta using periodic arrays of asymmetric pinning defects. *Phys. Rev. B* **68**, 014514 (2003).
26. Mishonov, T. M., Chéenne, N., Robbes, D. & Indekeu, J. O. Generation of 3rd and 5th harmonics in a thin superconducting film by temperature oscillations and isothermal nonlinear current response. *Eur. Phys. J. B* **26**, 291–296 (2002).
27. Ossandón, J. G. *et al.* Non-linear response of ac conductivity in narrow YBCO film strips at the superconducting transition. *J. Phys. Conf. Ser.* **43**, 655–658 (2006).
28. Foner, S. & McNiff, E. J. Upper critical fields of layered superconducting NbSe₂ at low temperature. *Phys. Lett. A* **45**, 429–430 (1973).
29. Xi, X. *et al.* Ising pairing in superconducting NbSe₂ atomic layers. *Nat. Phys.* **12**, 139–143 (2016).

Supplementary figures

1.

We added Fig. S4e and its figure caption.

2.

We added Figs. S8 and S9 along with their figure captions.

REVIEWER COMMENTS

Reviewer #1 (Remarks to the Author):

The manuscript entitled "Giant intrinsic rectification and nonlinear Hall effect under time-reversal symmetry in a trigonal superconductor" described a systematic experimental study on the nonlinear transport properties of PbTaSe₂ crystals discovered both in its normal state and in the superconducting state. The authors attributed the second-order nonlinear signal in the normal state to the anomalous scattering process and the enhanced nonlinearity at the superconducting state to the asymmetric Hall effect of vortex-anti-vortex string pairs in noncentrosymmetric system. Although rectification and nonlinear Hall effect have been reported in other material systems, the nonlinear effects that have been probed as excess voltage under time-reversal system has been rarely studied. Therefore, this paper is interesting to the readership of Nature Communications in this regard. However, there are two minor issues which need to be addressed in this manuscript before it is suitable to be published in Nature Communications.

1. According to the theoretical explanation, the rectification and nonlinear Hall effect essentially stem from noncentrosymmetric crystal structure of PbTaSe₂. Although the consistent results have been reproduced in seven PbTaSe₂ samples and in centrosymmetric NbSe₂ samples, it is more convincing and necessary to measure rectification effect with configuration B and nonlinear Hall effect with configuration A in the same PbTaSe₂ sample. The device with "sunbeam"-shaped electrodes can detect crystal anisotropy of nonlinear signals (Ref. Nat. Commun. 10(1): 1290; Nature, 547, 432-435 (2017)), which will be the best electrode geometry to meet the measurement requirement.

2. To comprehensively understand the results, the author should also provide basic information about the measured seven PbTaSe₂ samples in the supplementary materials, such as T_c, R-T curves, RRR, and mobilities.

Reviewer #2 (Remarks to the Author):

Authors have provided rather detailed reply to the criticism. Although I am not totally convinced that one needs to go beyond mean field theory to describe the rectification effect, the experimental results are interesting enough to be suitable for Nature Communications. I would like to ask authors to mention basic parameters of the considered superconducting material, such as e.g. London penetration length, coherence length at low temperatures, mean free path. I believe that this will help those who will try to develop more detailed theory of the observed effect.

Reviewer #3 (Remarks to the Author):

I appreciate the detailed response from the authors. I thought and still think the work is interesting and agree that a variety of second-order electrical responses can be used to measure the topological and geometrical properties of quantum materials.

However, I still believe the claim of a nonlinear Hall effect is not valid here. It is important to obey certain consensus of the field and not confuse the audience. As the authors mentioned, the effect (the skew scattering of chiral Bloch electrons in trigonal systems) they considered was theoretically predicted in Science Advances 6, eaay2497 (2020). The authors should have realized that the theory paper never called it a nonlinear Hall effect. The Science Advances paper consistently referred to this effect as rectification for both the longitudinal and transverse responses. This effect was never introduced as a nonlinear Hall effect throughout that paper.

In the reply, the authors also claimed that "What we report in the normal state of PbTaSe₂ in the

present manuscript can be considered to be the first experimental observation of this phenomenon." This is actually NOT true. In fact, it has already been reported in Nature Communications 12, 698 (2021) an electrical second-harmonic transverse response in Bi₂Se₃ with three-fold rotational symmetry, in which the Berry curvature dipole (BCD) induced nonlinear Hall effect is forbidden. The effect in Bi₂Se₃ was considered to arise dominantly from skew scattering in the topological surface states with its inherently chiral wave function, the same mechanism as the authors considered in the current manuscript. I want to emphasize that throughout the previous Nature Communications paper, the effect was referred to as frequency doubling or electrical SHG, NEVER nonlinear Hall.

In line with the careful claim in those previous papers, I insist on removing the claim of a nonlinear Hall effect in the current manuscript. Phenomenologically, yes, the observation is a second-order response in the transverse channel. However, the Hall effect has a unique physical meaning and is not a phenomenological definition. We can consider a case in the linear regime: in an orthorhombic lattice in which the resistivity along two principal axes is very different, if one injects current off the principal axes, a voltage perpendicular to the injected current direction can be generated without applying a magnetic field (RSI 88, 043901 (2017)). However, the transverse voltage generated in this scenario is not a Hall effect. It has the same mechanism as the longitudinal response.

As explained in my previous report, in PbTaSe₂ the second-order longitudinal and transverse response tensors are related by the symmetry of the system: $\rho_{xx} = -\rho_{yy}$. It means the longitudinal and transverse responses share the same mechanism. A few other theoretical papers (e.g., arXiv: 2106.12695) recently also distinguished a true nonlinear Hall effect (a unique mechanism to generate a Hall response, not longitudinal response) versus other second-order effects.

The authors should pay attention to the boundary between a nonlinear Hall effect and a second-order transverse response. The authors should also give credit to the work of Bi₂Se₃, which already showed a nonlinear transverse electrical response in a C₃ system without BCD. The novelty of the current paper compared to the previous one is the enhancement of the response in the superconducting regime. If the authors can reconsider these aspects, I recommend publication in Nature Communications.

Dear reviewers,

First of all, we thank the reviewers for carefully reading our manuscript and for constructive comments, which are all helpful to improve the scientific quality of the present manuscript. We have revised the manuscript so as to address the issues point by point.

We believe that the manuscript is now substantially improved and our findings would attract broad attention from audience working on superconductivity, topological material science, low dimensional materials, and any other symmetry-related physics.

Below, we provide a response and describe corrections.

In the main text, we highlighted all changes.

We would appreciate your kind consideration.

Sincerely,

Toshiya Ideue

Department of Applied Physics, University of Tokyo
7-3-1 Hongo, Bunkyo-ku, Tokyo, JAPAN 113-8656

Responses to Reviewers' comments

Reviewer #1:

The manuscript entitled “Giant intrinsic rectification and nonlinear Hall effect under time-reversal symmetry in a trigonal superconductor” described a systematic experimental study on the nonlinear transport properties of PbTaSe₂ crystals discovered both in its normal state and in the superconducting state. The authors attributed the second-order nonlinear signal in the normal state to the anomalous scattering process and the enhanced nonlinearity at the superconducting state to the asymmetric Hall effect of vortex-anti-vortex string pairs in noncentrosymmetric system. Although rectification and nonlinear Hall effect have been reported in other material systems, the nonlinear effects that have been probed as excess voltage under time-reversal system has been rarely studied. Therefore, this paper is interesting to the readership of *Nature Communications* in this regard. However, there are two minor issues which need to be addressed in this manuscript before it is suitable to be published in *Nature Communications*.

Response

We would appreciate reviewer #1 for her/his recognition of the importance of our study and her/his recommendation for publication in *Nature Communications*. Below, we have addressed the reviewer's comments with appropriate changes in the manuscript.

Comment 1

1. According to the theoretical explanation, the rectification and nonlinear Hall effect essentially stem from noncentrosymmetric crystal structure of PbTaSe₂. Although the consistent results have been reproduced in seven PbTaSe₂ samples and in centrosymmetric NbSe₂ samples, it is more convincing and necessary to measure rectification effect with configuration B and nonlinear Hall effect with configuration A in the same PbTaSe₂ sample. The device with “sunbeam”-shaped electrodes can detect crystal anisotropy of nonlinear signals (Ref. *Nat. Commun.* 10(1): 1290; *Nature*, 547, 432-435 (2017)), which will be the best electrode geometry to meet the measurement requirement.

Response

We agree that the device with sunbeam-shaped electrodes is beneficial to clarify the directional dependence of the nonlinear signals in one device. In order to observe the directional dependence of nonlinear transport with such a device, we need to etch the exfoliated flake into sunbeam shape and fabricate electrodes on it. Unfortunately, however, we found that our etching process damages the PbTaSe₂ flake, resulting in the disappearance of superconductivity and even insulating behavior. Thus, we conclude that preparing this type of device is difficult at this stage.

Although the suggested measurement is unfeasible, we believe that directional dependence of the nonlinear transport and reproducibility of the measurements has been clearly confirmed in the present results of multiple samples.

Comment 2

2. To comprehensively understand the results, the author should also provide basic information about the measured seven PbTaSe₂ samples in the supplementary materials, such as T_c , R - T curves, RRR , and mobilities.

Response

We thank reviewer #1 for this important suggestion. We showed the R - T curves of samples 3, 4, 5, 6, and 7, which are not shown in the main text, in Figs. S6a-e in Supplementary Information section V (Figs. R1a-e). Also, we added the values of T_c and RRR in each sample, to the Table S1 in Supplementary Information (Table R1). Although we did not measure Hall resistivity vs magnetic field curves in all samples, we plotted $\rho_{yx}^\omega(B)$ of sample 6 in Fig. S6f in Supplementary Information (Figs. R1f) and analyzed it by using the two-carrier model (yellow dashed line)

$$\rho_{yx}^\omega(B) = \frac{B}{e} \frac{(n_h \mu_h^2 - n_e \mu_e^2) + \mu_h^2 \mu_e^2 (n_h - n_e) B^2}{(n_h \mu_h + n_e \mu_e)^2 + \mu_h^2 \mu_e^2 (n_h - n_e)^2 B^2}$$

where n_h (n_e) and μ_h (μ_e) are the carrier density and mobility of holes (electrons), respectively. Obtained fitting parameters are $n_h = 5.4 \times 10^{22} \text{ cm}^{-3}$, $n_e = 4.8 \times 10^{22} \text{ cm}^{-3}$, $\mu_h = 4.7 \times 10^4 \text{ cm}^2 \text{V}^{-1} \text{s}^{-1}$, and $\mu_e = 5.0 \times 10^4 \text{ cm}^2 \text{V}^{-1} \text{s}^{-1}$. These values are in the same order as ones in sample 5 shown in the Supplementary Information section I. We added the above information in Supplementary Information section V.

Figure R1. Basic properties of measured samples. a-e, Temperature dependences of the first harmonic resistance R_{xx}^{ω} in samples 3, 4, 5, 6, and 7. f, Transverse resistivity ρ_{xx}^{ω} at $T = 5$ K in sample 6. Yellow dashed line shows the fitting curve for $\rho_{yx}^{\omega}(B)$ by the two-carrier model.

Sample No.	configuration	R_{xx}^{ω} (m Ω) ($T = 5$ K)	RRR	T_c (K)	$R_{xx}^{2\omega}$ (m Ω) (normal)	$R_{yx}^{2\omega}$ (m Ω) (normal)	$R_{xx}^{2\omega}$ (m Ω) (SC)	$R_{yx}^{2\omega}$ (m Ω) (SC)
1	A	200	14	3.09	0.09 (20 K)	1.4 (20 K)	0.76 (150 μ A)	2.4 (150 μ A)
2	B	48	95	3.61	0.22 (50 K)	-0.052 (50 K)	1.0 (520 μ A)	-0.34 (520 μ A)
3	A	47	40	3.66	0.1 (20 K)	0.61 (20 K)	-0.4 (170 μ A)	3.8 (170 μ A)
4	B	28	108	3.56	1.4 (50 K)	0.024 (50 K)	1.1 (80 μ A)	-0.030 (80 μ A)
5	B	87	79	3.68	0.57 (20 K)	-0.11 (20 K)	4.8 (90 μ A)	1.8 (90 μ A)
6	A	110	46	3.73	0.10 (20 K)	0.22 (20 K)	0.34 (310 μ A)	1.6 (310 μ A)
7	B	180	29	3.69	0.52 (50 K)	-0.05 (50 K)	5.8 (270 μ A)	-0.78 (270 μ A)

Table R1. Summary of the nonlinear anomalous transport in all samples. We added the values of RRR and T_c .

Reviewer #2:

Authors have provided rather detailed reply to the criticism. Although I am not totally convinced that one needs to go beyond mean field theory to describe the rectification effect, the experimental results are interesting enough to be suitable for *Nature Communications*. I would like to ask authors to mention basic parameters of the considered superconducting material, such as e.g. London penetration length, coherence length at low temperatures, mean free path. I believe that this will help those who will try to develop more detailed theory of the observed effect.

Response

We would appreciate reviewer #2 for her/his recognition of the importance of our study and her/his recommendation for publication in *Nature Communications*.

Following the suggestion by Reviewer #2, we mentioned the basic parameters of PbTaSe₂ in Supplementary Information section V. In-plane London length at $T = 0$ K is $\lambda_{ab}(0) = 82$ nm and coherence length at $T = 0$ K is $\xi_{ab}(0) = 41$ nm (*Science of Advanced Materials* **8**, 2097 (2016)). We also estimated mean free path ℓ as follows. Because PbTaSe₂ is a semimetal with multi-carrier behavior and the main carrier is the hole, we roughly estimate ℓ as $\ell \sim \frac{\hbar k_F}{n_h e^2 \rho_{xx}^\omega} \sim 69$ nm, where $k_F \sim 0.5 \text{ \AA}^{-1}$ is the Fermi wavenumber of the hole pocket (*Nature Communications* **7**, 10556 (2016)), $n_h = 5.7 \times 10^{22} \text{ cm}^{-3}$ is the carrier density of holes in sample 5, $\rho_{xx}^\omega = 0.53 \text{ } \mu\Omega\text{cm}$ is the longitudinal resistivity in sample 5.

Reviewer #3:

I appreciate the detailed response from the authors. I thought and still think the work is interesting and agree that a variety of second-order electrical responses can be used to measure the topological and geometrical properties of quantum materials.

Response

We would appreciate reviewer #3 for her/his recognition of the significance of our study. Below, we have addressed the reviewer's comments with appropriate changes in the manuscript.

Comment 1

However, I still believe the claim of a nonlinear Hall effect is not valid here. It is important to obey certain consensus of the field and not confuse the audience. As the authors mentioned, the effect (the skew scattering of chiral Bloch electrons in trigonal systems) they considered was theoretically predicted in Science Advances 6, eaay2497 (2020). The authors should have realized that the theory paper never called it a nonlinear Hall effect. The Science Advances paper consistently referred to this effect as rectification for both the longitudinal and transverse responses. This effect was never introduced as a nonlinear Hall effect throughout that paper.

In the reply, the authors also claimed that "What we report in the normal state of PbTaSe₂ in the present manuscript can be considered to be the first experimental observation of this phenomenon." This is actually NOT true. In fact, it has already been reported in Nature Communications 12, 698 (2021) an electrical second-harmonic transverse response in Bi₂Se₃ with three-fold rotational symmetry, in which the Berry curvature dipole (BCD) induced nonlinear Hall effect is forbidden. The effect in Bi₂Se₃ was considered to arise dominantly from skew scattering in the topological surface states with its inherently chiral wave function, the same mechanism as the authors considered in the current manuscript. I want to emphasize that throughout the previous Nature Communications paper, the effect was referred to as frequency doubling or electrical SHG, NEVER nonlinear Hall.

In line with the careful claim in those previous papers, I insist on removing the claim of a nonlinear Hall effect in the current manuscript. Phenomenologically, yes, the observation is a second-order response in the transverse channel. However, the Hall effect has a unique physical meaning and is not a phenomenological definition. We can consider a case in the linear regime: in an orthorhombic lattice in which the resistivity along two principal axes is very different, if one injects current off the principal axes, a voltage perpendicular to the injected current direction can be generated without applying a magnetic field (RSI 88, 043901 (2017)). However, the transverse voltage generated in this scenario is not a Hall effect. It has the same mechanism as the longitudinal response.

As explained in my previous report, in PbTaSe₂ the second-order longitudinal and transverse response tensors are related by the symmetry of the system: $\rho_{xx} = -\rho_{yy}$. It means the longitudinal and transverse responses share the same mechanism. A few other theoretical papers (e.g., arXiv: 2106.12695) recently also distinguished a true nonlinear Hall effect (a unique mechanism to generate a Hall response, not longitudinal response) versus other second-order effects.

The authors should pay attention to the boundary between a nonlinear Hall effect and a second-order transverse response. The authors should also give credit to the work of Bi₂Se₃, which already showed a nonlinear transverse electrical response in a C₃ system without BCD. The novelty of the current paper compared to the previous one is the enhancement of the response in the superconducting regime. If the authors can reconsider these aspects, I recommend publication in Nature Communications.

Response

We understand that it remains an open question whether the nonlinear transverse response in trigonal crystals can be called nonlinear Hall effect or not. Although the original theory paper (Science Advances 6, eaay2497 (2020)), which predicts the nonlinear longitudinal and transverse responses in trigonal systems, never called it a nonlinear Hall effect as Reviewer #3 mentioned, we would like to point out that other theory papers and recent review papers of the nonlinear Hall effect (*Nat. Commun.* **10**, 3047 (2019), *Nat. Commun.* **12**, 5038 (2021), *Nature Review Physics* **3**, 744 (2021) and *Nature Materials* **20**, 1604 (2021)) include this scattering induced nonlinear response as nonlinear Hall effect. In view of all these controversial situations, we decided to use “nonlinear transverse response” for the present phenomenon instead of “nonlinear Hall effect” to avoid the confusion.

Also, we would thank reviewer #3 for informing us the recent paper (*Nature Communications* **12**, 698 (2021)), which reported the nonlinear transverse response at the trigonal surface of Bi₂Se₃. We cited this paper in the main text and compared the values of nonlinear signals with our results of PbTaSe₂ in Fig. 4b (Fig. R2). Interestingly, the data of Bi₂Se₃ are also aligned in the yellow region in this plot, showing the similar behavior to those of few-layer WTe₂, bulk TaIrTe₄, and PbTaSe₂. This may reveal the universal feature of the scattering-induced nonlinear transverse response. It should be noted that the PbTaSe₂ shows the largest values of the nonlinear transport. In addition, we would like to emphasize that present work is the first report of the nonlinear longitudinal and transverse response under time reversal symmetry in superconducting region, which will not only accelerate the search of new types of nonlinear transport in quantum materials but also pave the way for investigating novel functionalities in noncentrosymmetric conductors.

Figure R2. Summary of nonlinear transverse signals in several kinds of samples We added the data for Bi_2Se_3 surface extracted from *Nature Communications* **12**, 698 (2021).

List of Revisions

Main text

1.

We changed the title from

“Giant intrinsic rectification and nonlinear Hall effect under time-reversal symmetry in a trigonal superconductor”

to **“Giant second harmonic transport under time-reversal symmetry in a trigonal superconductor”**

2.

We revised all the expressions of “nonlinear anomalous Hall effect” regarding to our results.

3.

We added subsections.

4. Line 110, page 5

We revised from

“Note that the intrinsic rectification effect (Fig. 2c), which has never been reported under time-reversal symmetry, is clearly observed as well as the nonlinear anomalous Hall effect (Fig. 2b).”

to **“We note that similar nonlinear transverse response has been reported in the trigonal surface of Bi₂Se₃²⁵. In the present PbTaSe₂, the intrinsic rectification effect (Fig. 2c), which has never been reported under time-reversal symmetry, is also clearly observed as well as the nonlinear transverse response (Fig. 2b).”**

5. Line 240, page 11

We added

“In Bi₂Se₃ surface, skew scattering is the main origin of the nonlinear transverse response since Berry curvature dipole is absent in trigonal symmetry²⁵.”

6. Line 243, page 11

We revised from

“ $10^{-3} \sim 10^{-2} \mu\text{mV}^{-1}$ ”

to **“ $10^{-4} \sim 10^{-2} \mu\text{mV}^{-1}$ ”**.

7. Line 244, page 11

We revised from

“WTe₂”

to “**other materials**”.

8. Line 246, page 11

We added

“**Bi₂Se₃ surface**”.

References

We added the following reference.

25. He, P. *et al.* Quantum frequency doubling in the topological insulator Bi₂Se₃. *Nat. Commun.* **12**, 698 (2021).

Figures

1. Line 492, page 24

We added

“**Bi₂Se₃ surface**”.

2. Line 495, page 24

We added

“**Bi₂Se₃ surface**”.

3. Line 502, page 25

We added

“**Gray triangles depict $\frac{|E_y^{(2)}|}{(E_x^{(1)})^2}$ in the Bi₂Se₃ surface. Data were sourced from He et al.²⁵.”.**

4.

We replaced Figure 4b to a new figure.

Supplementary Information

1.

We revised all the expressions of “nonlinear anomalous Hall effect” regarding to our results.

2. Line 336, page 20

We added

”Here, we show basic parameters of PbTaSe₂. We show temperature versus linear resistance R_{xx}^ω curves in samples 3, 4, 5, 6, and 7, which are not shown in the main text, in Figs. S6a-e. The values of residual resistivity ratio $RRR = R_{xx}^\omega(T = 300 \text{ K})/R_{xx}^\omega(T = 2 \text{ K})$ and transition temperature T_c are summarized in Table S1. In Fig. S6f, we plotted $\rho_{yx}^\omega(B)$

in sample 6 and analyzed it by using the two-carrier model (yellow dashed line) similarly to the analysis in the Supplementary Information section I. Obtained fitting parameters are $n_h = 5.4 \times 10^{22} \text{ cm}^{-3}$, $n_e = 4.8 \times 10^{22} \text{ cm}^{-3}$, $\mu_h = 4.7 \times 10^4 \text{ cm}^2\text{V}^{-1}\text{s}^{-1}$, and $\mu_e = 5.0 \times 10^4 \text{ cm}^2\text{V}^{-1}\text{s}^{-1}$. These values are in the same order as those in sample 5 shown in the Supplementary Information section I. We also note the characteristic parameters in superconducting state of PbTaSe₂: in-plane London length at $T = 0 \text{ K}$ is $\lambda_{ab}(0) = 82 \text{ nm}$ and in-plane coherence length at $T = 0 \text{ K}$ is $\xi_{ab}(0) = 41 \text{ nm}$ ¹³. We estimated mean free path ℓ as follows. Because PbTaSe₂ is a semimetal with multi-carrier behavior and the main carrier is the hole, we roughly estimate ℓ as $\ell \sim \frac{\hbar k_F}{n_h e^2 \rho_{xx}^\omega} \sim 69 \text{ nm}$, where $k_F \sim 0.5 \text{ \AA}^{-1}$ is the Fermi wavenumber of the hole pocket²⁶, $n_h = 5.7 \times 10^{22} \text{ cm}^{-3}$ is the carrier density of holes in sample 5, $\rho_{xx}^\omega = 0.53 \text{ } \mu\Omega\text{cm}$ is the longitudinal resistivity in sample 5.”

Supplementary references

We added the following reference.

26. Bian, G. *et al.* Topological nodal-line fermions in spin-orbit metal PbTaSe₂. *Nat. Commun.* **7**, 10556 (2016).

Supplementary figures

1.

We revised Table S1.

2.

We added Fig. S6 along with its figure caption.